# Hard Successive Interference Cancellation for M-QAM MIMO Links in the Presence of Rayleigh Deep-Fading

**DOI:** 10.3390/s24155038

**Published:** 2024-08-03

**Authors:** Avner Elgam, Meir Klemfner, Shachar Silon, Yossi Peretz, Yosef Pinhasi

**Affiliations:** 1Faculty of Engineering, Ariel University, Ariel 40700, Israel; yosip@ariel.ac.il; 2Department of Computer Sciences, Lev Academic Center Jerusalem, College of Technology, Jerusalem 9372115, Israel; mklefne@g.jct.ac.il (M.K.); silon@g.jct.ac.il (S.S.); yosip@g.jct.ac.il (Y.P.)

**Keywords:** M-QAM, MIMO, Rayleigh deep-fading, interference cancellation, parallel decoding, computational feedback

## Abstract

In our paper, we propose a generalized version of the Alternating Projections Digital Hard Successive Interference Cancellation (AP-HSIC) algorithm that is capable of decoding any order of constellation M in an M-Quadrature Amplitude Modulation (QAM) system. Our approach applies to Rayleigh deep-fading Multiple-Input Multiple-Output (MIMO) channels with high-level Additive White Gaussian Noise (AWGN). It can handle various destructive phenomena without restricting the number of antenna arrays in the transmitter/receiver. Importantly, it does not rely on closed-loop MIMO feedback or the need for Channel-State Information Transmission (CSIT). We have demonstrated the effectiveness of our approach and provided a Bit Error Rate (BER) analysis for 16-, 32-, and 64-QAM modulation systems. Real-time simulations showcase the differences and advantages of our proposed algorithm compared to the Multi-Group Space-Time Coding (MGSTC) decoding algorithm and the Lagrange Multipliers Hard Successive Interference Cancellation (LM-HSIC) algorithm, which we have also developed here. Additionally, our paper includes a mathematical analysis of the LM-HSIC algorithm. The AP-HSIC algorithm is not only effective and fast in decoding, including interference cancellation computational feedback, but it can also be integrated with any Linear Processing Complex Orthogonal Design (LPCOD) technique, including Complex Orthogonal Design (COD) schemes such as high-order Orthogonal Space–Time Block Code (OSTBC) with high-order QAM symbols.

## 1. Introduction

The field of advanced wireless MIMO communication technology is rapidly developing and has attracted significant research interest. Current MIMO technology has greatly improved in terms of reliability, effectively decoding ultra-high transmission bit rates with minimal performance loss and decoding heavy information based on high-order QAM constellations with high BER performance [1,2,3]. These systems’ performance and features are achieved thanks to mechanisms that enable the creation of spatial-multiplexing and uncorrelated MIMO channels with high diversity order, high spatial selectivity, and diversity gain. Other significant advantages of modern wireless MIMO systems include their effectiveness in dealing with various interferences on the wireless MIMO channel, such as Rayleigh deep-fading [4,5,6], multi-path effects [3,7], and scattering phenomena [8,9], the fast scale of MIMO fading [10], and dealing with multiple interference phenomena and jamming attacks [11] with several methods of interference cancellation [2,10,12,13]. These capabilities were possible thanks to the intelligent utilization of the spatial domain [14]. The requirements for the application of advanced engineering in wireless MIMO networks, such as Dynamic and Shared Spectrum Access (DSSA) technology  [15], adaptive equalizers, and modern estimation techniques, provide stability in the decoding process and an accurate estimation of high-order constellations under bad propagation conditions, and make up the basic construction of traditional networks [1] (e.g., 5th Generation New Radio (5G-NR) Heterogeneous Network (HetNet) [15,16], 6th-mobile-generation [17], and Wi-Fi-6 802.11.ax). These requirements, in addition to achieving Ultra-Reliable Low-Latency (URLL) in the decoding process, including the capability of digital Successive Interference Cancellation (SIC) and the optimization of the spectrum access, are also needed, especially in Unmanned Aerial Vehicle (UAV) communications [10], point-to-point Stand-Alone (SA) wireless MIMO network systems [10], communication between autonomous vehicles [18], tactical MIMO communication systems in the military field, communication medical equipment, and many more.

Strengthening the cognitive intelligence of the wireless MIMO communication system’s backbone is a critical contribution to developing modern wireless MIMO systems [19,20,21]. One of the key features of intelligent mechanisms is the use of parallel spatial decoding algorithms that are based on real-time hard decisions [22,23]. Spatial decoding algorithms that include features, such as high reliability, robustness, and flexibility, are creating conditions that allow a paradigm in which we are freed from constraints as much as possible (e.g., without the need for physical feedback between the receiver and the transmitter, enabling us to share the CSIT [7,10,24]).

Our research is focused on developing a spatial decoding algorithm and studying destructive phenomena in the wireless channel. The wireless channel plays a crucial role in the transmission process, and it is important for researchers to develop effective methods for quickly and accurately processing MIMO wireless channel matrices, which represent the “propagation characteristics”. These methods and studies are essential for the development of advanced communication systems. A good communication channel enables optimal throughput, spectral efficiency, and network capacity and minimizes power consumption. In advanced MIMO wireless communication systems, adjusting the channel parameters to support next-generation technologies can result in favorable propagation characteristics, especially in higher frequency bands [25]. Current approaches such as massive MIMO, Coordinated Multipoint Operation (CoMP), Millimeter Wave, and Industrial Internet of Things (IIoT) rely on higher frequency bands and contribute to advancements beyond 5G-NR or 6th-mobile-generation requirements, also known as next-generation technologies [26].

Our research aligns with the trends and goals of deploying these advanced technologies, particularly in urban environments, indoor applications, large and dense cities, and areas with many end users. This aims to significantly improve real-time connectivity and drastically increase information rates. The communication bottleneck depends on the wireless medium [27]. All these areas that we mentioned are characterized by Rayleigh deep-fading phenomena, which damage the channel significantly [28]. Therefore, it is crucial to integrate spatial SIC decoding algorithms at the front end of the receiver component of advanced base stations, Access Points (APs), relays, and advanced modems. For example, in deploying small-cell HetNet in 5G-NR networks, these components play a critical role in increasing the number of users and the channel’s capacity in a given area. However, the presence of Rayleigh deep-fading phenomena between the relay and the end users may cause the scheduler to lower the order of the transmitted constellations. Thus, integrating spatial SIC decoding algorithms such as AP-HSIC or LM-HSIC in the Radio Intelligence Controlling (RIC) of the small-cell HetNet is significant. These algorithms can counteract destructive phenomena and determine high-order constellation orders.

Few spatial SIC decoding algorithms constitute exemplary contributions, such as the possibility of an extension approach in the modulation order and the generality of time–frequency–space scenario solutions via the decoding core mechanisms. These approaches apply to all modern communication systems’ high-quality operation. The meaning of these concepts pertains to creating a situation where the algorithmic complexity overheads and the computational complexity overheads of the decoding algorithms are preserved low, even though the order of the QAM constellation increases significantly. The existence of the convergence capability of the decoding algorithm and maintaining the generalization principle must be taken into account, even though the order of the constellation increases significantly [1,22,29]. For example, a decoding algorithm developed for 16-QAM must be able to decode constellations of significantly higher orders of QAM while maintaining properties of generalization, fast convergence, and reasonable computational complexity [1,5,22,29]. In addition, the utilization of any number of multiple antenna arrays in the transmitter as well as in the receiver side without any computational limitation, such as power domain multiplexing management, for example, is essential for advanced MIMO schemes such as massive-MIMO systems [30].

Another example of the essential contribution of a few advanced spatial SIC decoding algorithms is to allow, in the development process of the algorithms, the approach of the accurate decoding of any value of constellation order or any arbitrary mapping constellations of bits to the M-QAM symbol, especially in a high order of QAM [5], as we mentioned, but under challenging interference scenarios. In addition, the intelligent development of advanced spatial SIC decoding algorithms allows integration with any version of COD scheme approaches such as OSTBC [7] or Orthogonal Space-Frequency Block Codes (OSFBC) [7]. As we mentioned, we focus on the ability to overcome different types of interference and deal with a diversity of interference phenomena, such as wireless destruction channels, small/large scale fading, deep-fading, and interference jamming  [31,32,33,34,35,36], with Channel-State Information Receivers (CSIRs) only.

Clearly, to cope with these challenges, several types of advanced algorithms have been developed and are increasingly embedded in many wireless MIMO communication systems, including spatial selectivity capability and breakthrough techniques (e.g., high-resolution beam-forming techniques  [37,38], hybrid beam-forming  [39,40], classical digital SIC [10], and non-conventional beam-forming based on Machine Learning (ML) digital SIC [41,42], Non-Orthogonal Multiple Access (NOMA) algorithms  [43,44], and several algorithms for massive MIMO systems  [30,45,46]).

In recent years, these advanced techniques have combined several linear and non-linear detection algorithms that try to achieve optimal performance in aspects of low computational complexity, fast rate of convergence, the minimization of the BER, and maximizing the channel’s capacity. Examples of using traditional linear and non-linear detector algorithms include Linear Minimum Mean-Square-Error Interference Cancellation (LMMSE-IC) [7] detector, Zero Forcing Interference Cancellation (ZF-IC) [7], Minimum Mean-Square-Error Interference Rejection Combined (MMSE-IRC)  [2,47,48,49], Message Passing Detection (MPD) [1] method, Low Complexity Message Passing Detection (LCMPD) algorithms [29], Log-Likelihood Ratio (LLR) algorithms [5,50], Monte Carlo sampling techniques (e.g., Markov Chain Monte Carlo (MCMC) algorithms [51]), and the MGSTC algorithm [7,52].

The common denominator of all these spatial SIC decoding algorithms is to focus on the ability to decode very quickly and with a high reliability and ultra-fast information rate based on the high order of QAM constellations while maintaining the complexity and low computational cost, all under the assumptions of MIMO Rayleigh fading channels. On one hand, all these algorithms achieve near-optimal performance and a high order of diversity gain with high BER performance. On the other hand, these techniques assume several fundamental assumptions, the non-existence of which in the real-physical environment of the MIMO wireless may cause significant performance degradation and even collapse (i.e., non-convergence or high BER levels) of the algorithms. Examples of basic assumptions in constructing the MPD and LCMPD are that the wireless matrix of the MIMO channels is under the hardening channel model and that its component values do not change throughout the channel estimation and the decoding process [1,29]. In algorithms of a serial nature [1,7,29,42,52,53,54], these basic assumptions are critical since for any significant disturbance or distortion that was not taken into account in the assumed model, the serial process will lead to the amplification of computational errors and even to the collapse of the technique. Moreover, in the fast-scale medium, the assumption of the hardening channel model is not realistic [10,45].

Advanced spatial SIC decoding algorithms must address the identity and nature of interference in space in wireless communication reality [3]. Rayleigh deep-fading is a prime example of a challenging interference phenomenon [4,55]. Rayleigh deep-fading is one of the most complex and destructive phenomena in the MIMO wireless communication topics [4,55]. This phenomenon occurs when objects with a reflective and dispersive nature, such as scatters, are located in close proximity to the antenna arrays on the transmitter/receiver side, causing self-interference between the antenna array. Rayleigh deep-fading also causes a high spatial correlation between the antennas in the array. The high-correlation effect destroys the orthogonality between the columns of the wireless MIMO channel matrix and hits on the spatial-multiplexing capabilities between the transmitting–receiving signals [4,7,55]. The likelihood of experiencing deep-fade in wireless channels is significantly higher in the lower region of the Probability Density Function (PDF) of the Rayleigh distribution, on average. In many cases, the Rayleigh deep-fading phenomenon significantly decreases the BER performance and increases the number of decoding iterations of the algorithms. All the algorithms we have reviewed so far in this article do not handle the Rayleigh deep-fading scenarios.

Another critical aspect that advanced decoding algorithms such as LCMPD, MPD, and LLR must focus on and treat is the exact identification of the statistical distribution of the interference and the noise parts. These algorithms generally establish a basic assumption that the noise components received in the receiver and the matrix of transmitted symbols have a Gaussian distribution [1,29]. This reference makes it possible to calculate the interference component’s duration and variability and estimate the conditional probability function of editing a transmitted symbol matrix given the channel matrix and the received signal matrix in the decoding process. Therefore, if there is an interference with a non-Gaussian distribution or a channel medium that causes the distribution of the transmitted signal to change [15,45], it will create many errors in the calculation processes of these algorithms.

This paper proposes a generalized version of the new spatial SIC decoding algorithms—the AP-HSIC [56]—that only requires the CSIR assumption and generates computational feedback at the receiver. This computational feedback can overcome the effects of random scatters in a multi-path fading channel and several interference effects, such as Rayleigh deep-fading under quasi-static flat Rayleigh fading MIMO channels combined with high-leveled AWGN. In this paper, we present the algorithm’s BER performance of three systems based on three spatial SIC decoding algorithms and compare them. The first system, the MGSTC spatial SIC decoding algorithm, is based on the MIMO array’s serial-decoding mode. The second system is based on the AP-HSIC algorithm applied to *M*-QAM, which combines parallel processing decoding methods in the MIMO array. The last system is the proposed LM-HSIC algorithm, based on Lagrange multipliers optimization and parallel decoding. This comparison was conducted in an environment of different Rayleigh deep-fading, high-level AWGN, and scatter scenarios. The comparison is reflected in the performance levels of BER vs. the average Signal-to-Noise Ratio (SNR). The analysis considers different high constellation orders (16, 32, and 64–QAM) in the three wireless system networks (MGSTC, AP-HSIC, LM-HSIC). We also present the theoretical graphs of the BER vs. SNR of the three constellation orders with AWGN only and the BER vs. SNR graphs of the three constellation orders with fading without an error correction mechanism at the receiver. We compared all these graphs with the BER vs. SNR graphs of the three constellation orders with the AP-HSIC performance under fading scenarios.

The proposed AP-HSIC and LM-HSIC algorithms for M-QAM offer an advanced solution for effectively canceling digital interference. This innovative approach provides unprecedented flexibility for SA networks without the need to control the channel or share a channel that statistically measures the spatial domain. In addition, the proposed algorithms enable the real-time decoding of symbol matrices in a parallel MIMO mode, ensuring seamless interference cancellation without compromising the decoding rate. These capabilities are a game-changer for ultra-high bit-rate and high-capacity channel requirements. With the proposed algorithms, we can differentiate between the original channel–response matrix and interfering factors, ΔH, allowing for the accurate calculation and timely updating of the general MIMO channel–response matrix online. The computation of ΔH allows a total offset of the interference/deep-fading effects. As a result, the system can decode the original symbols without additional transmission power consumption or the need for re-transmission. Furthermore, the statistical dependencies between general interference (or jamming signal) and the user signals, as well as the AWGN and the user signals, further enhance the system’s effectiveness (see Lemma F.1 in [56]). This represents a pivotal advancement in addressing interference and Rayleigh deep-fading effects.

The remaining sections of the paper are as follows: Section 2 presents the proposed communication MIMO model under Rayleigh deep-fading scenarios. Section 3 introduces the changes in the new version of the AP-HSIC algorithm. Section 4 provides a description of the LM-HSIC algorithm, and Section 5 details the real-time simulations and numerical results based on SIMULINK and MATLAB platforms, offering a comparison of the performances of three methods: the MGSTC, the AP-HSIC, and the LM-HSIC. In Section 6, we draw conclusions and present a vision for further research on efficient solutions to communication system bottleneck problems under interference scenarios.

## 2. Proposed Communication MIMO Model under Rayleigh Deep-Fading MIMO Channels with a High Order of QAM-Constellation Decoding Capability

This section describes a proposed communication MIMO model for two methods under various Rayleigh deep-fading scenarios. Deep-fading correlation significantly impacts the performance of multiple-antenna systems, especially in the adaptive digital signal process and the decoding process [4,55]. The double-correlated Rayleigh deep-fading channels are present on both sides of the system, on the transmitter and receiver sides, as was studied in  [4]. Estimating Rayleigh deep-fading and improving the system’s performance is one of the most significant challenges in wireless communication systems [4,10,55].

In this section, we describe and analyze two methods of wireless communication MIMO systems. The MIMO model assumes Rayleigh deep-fading MIMO channels, and in Section 5, we compare the BER performance aspects of these two architecture methods. The first method uses a transmitter based on the MGSTC scheme  [7,52]. It incorporates the diversity-transmitting technique known as OSTBC. The receiver employs the MGSTC-SIC decoding algorithm, as detailed in  [7,52]. The MGSTC scheme is capable of interference cancellation and decoding, allowing it to extract a single symbol information block from a group of transmitted blocks and merge them using an error-reducing mechanism. This technique operates by iteratively decoding the data at the receiver. It involves the parallel transmission of a sequence of blocks in space and, at the receiver, a serial decoding process (i.e., multi-stage decoding), which separates the different transmitted blocks. The MGSTC decoding algorithm decodes and separates the individual data streams from a specific user from the other streams. Multiple-stage decoding creates a communication mode that can mitigate the impact of noise, such as the sum of AWGN with the other broadcast blocks. This is achieved by leveraging the accuracy of a single decoding stream in relation to the previous decoding iteration.

The primary advantage of the MGSTC decoding algorithm is that at the end of each iteration when we move to the next group order, it provides a diversity gain for every matrix of OSTBC transmission modulation symbols, $S_{c_{i}}$ (MGSTC includes *I* component-blocks $S_{c_{i}}$, for group $c_{i}$, i=1,...,I). The diversity gain is increased by $n_{i}\times \left(n_{1}+...+n_{i}+N_{r}-N_{t}\right)$ when *i* is the iteration number and $n_{i}$ is the number of transmission antennas in the *i*’th group, while $N_{r}$, and $N_{t}$ are the number of total receiver antennas and the number of total transmission antennas, respectively.

The challenge with MGSTC and multi-stage decoding lies in the fact that the benefits in certain communication scenarios, such as when spatial interference occurs, can become drawbacks. The drawbacks are evident in the SIC process of the MGSTC, as it operates in a serial decoding mode for data blocks. Any errors accumulated in a given SIC process are repeated to the next iteration and amplified. Another notable weakness of the MGSTC is the wireless links between the antenna pairs in the descending order of the transmitter’s antenna array (e.g., antennas 5 and 6 in our simulation) to all antenna arrays of the receiver. These paths suffer more attenuation regarding the SNR or Signal-Interference Ratio (SIR) and rely on diversity gain. Consequently, when facing interfering signals, deep-fading effects, or high levels of AWGN, an imbalance in the trade-off between higher diversity gain and a lower SNR occurs, leading to increased values of the BER. The second method we propose is divided into two sub-approaches. The first sub-approach uses the MGSTC-OSTBC at the transmitter and the AP-HSIC decoding algorithm at the receiver instead of the MGSTC decoding algorithm in the first method. In the second sub-approach, we use the MGSTC-OSTBC at the transmitter and the LM-HSIC at the receiver. This hybrid scheme provides immunity against various interference scenarios, as shown in the simulation results below and in  [56]. Additionally, it achieves a more efficient and accurate process of parallel spatial decoding of the series of simultaneously transmitted symbol blocks compared to the first scheme we described. We separately analyze and simulate the two methods under challenging interference Rayleigh deep-fading scenarios in the same spatial domain.

It is important to emphasize that in both methods, we simultaneously transmit all the transmission sub-blocks, $S={\left[S_{c_{1}}^{T},S_{c_{2}}^{T},...,S_{c_{I}}^{T}\right]}^{T}$, in parallel. However, in the first MGSTC method, the MGSTC decoding algorithm decodes the sub-blocks serially, while in the second method that we propose, the receiver, based on the AP-HSIC and on the LM-HSIC algorithms, decodes all the sub-blocks at the same time, based on parallel iterations of offsetting the noise/interfering part in the channel, as we will present later.

We start the first and essential analysis with the Rayleigh deep-fading case. In that case, the communication standard model, $Y=\sqrt{P}HS+Z$ changes (see [4]):(1)$$Y=\sqrt{P}H_{TR}S+\sqrt{\frac{P}{SNR}}Z,$$
where
(2)$$H_{TR}=R^{\frac{1}{2}}HT^{\frac{1}{2}}$$
where, essentially, the Kronecker correlation structure of the MIMO–transmission–receiver matrix is $H_{TR}$ with dimension $\left[N_{r}\times N_{t}\right]$ [4,55]. The elements of *H* are the conventional MIMO channel matrix, where *R* and *T* are the receiving and transmitting Rayleigh deep-fading self-correlation matrices. Let
(3)$$\left\{\begin{array}{c} \Delta T=I-T^{\frac{1}{2}}\\ \Delta R=I-R^{\frac{1}{2}} \end{array}\right.$$
where ΔT and ΔR are the changes in the Rayleigh deep-fading self-correlation matrices on the transmitter and receiver sides, respectively, concerning the matrix *H*. Next, *Y* is the received signal matrix, with dimensions $\left[N_{r}\times k\right]$, where *k* is the number of sample symbols per frame and *S* is the matrix of OSTBC transmission modulation symbols of the desired transmission, with dimensions $\left[N_{t}\times k\right]$. Finally, the SNR is the signal-to-noise ratio, *P* is the total transmission power, and *Z* is the independent complex Gaussian random variable noise.

We can rewrite (Equation 2) by placing (Equation 3) into (Equation 2) to obtain the following:(4)$$\begin{array}{cc} H_{TR} & =\left(I-\Delta R\right)H\left(I-\Delta T\right)\\ \; & =H-\Delta RH-H\Delta T+\Delta RH\Delta T\\ \; & =H+\Delta H, \end{array}$$
where
(5)ΔH=−ΔRH−HΔT+ΔRHΔT.

The new form of the MIMO model is, therefore,
(6)$$Y=\sqrt{P}\left(H+\Delta H\right)S+\sqrt{\frac{P}{SNR}}Z.$$

In the following, we would write
(7)Y=(H+ΔH)S+Z,
where the normalizing factors $\sqrt{P},\sqrt{\frac{P}{SNR}}$ were suppressed into the related variable for convenience. Taking the expectation on both sides of (Equation 7), we obtain the following:(8)$$E\left[Y\right]=E\left[H+\Delta H\right]E\left[S\right],$$
since, by assumption, $E\left[Z\right]=0$ and since *S* and H+ΔH are assumed to be statistically independent. In the sequel, we will therefore write the following:(9)$$Y=\left(H+\Delta H\right)S,$$
for convenience.

To demonstrate the weakness in the MGSTC serial decoding algorithm in the presence of the Rayleigh deep-fading phenomenon, the equation that presents the first iteration  [7], where ${\tilde{Y}}_{c_{1}}$ is the received signal matrix for decoding the first symbol block $S_{c_{1}}$, is as follows:(10)$$\begin{array}{cc} {\tilde{Y}}_{c_{1}} & =\Theta_{c-c_{1}}Y=\Theta_{c-c_{1}}\left[\sqrt{P}HS+\sqrt{P}\Delta HS+\sqrt{\frac{P}{SNR}}Z\right]\\ \; & =\sqrt{P}\Theta_{c-c_{1}}H_{c_{1}}S_{c_{1}}+\sqrt{P}\Theta_{c-c_{1}}\Delta HS+\sqrt{\frac{P}{SNR}}{\tilde{Z}}_{c-c_{1}}, \end{array}$$
where $H=\left[\begin{array}{cccc} H_{c_{1}} & H_{c_{2}} & \cdots & H_{c_{I}} \end{array}\right]$ and $\Theta_{c-c_{1}}$ is the null-space matrix relative to the decoding process of $S_{c_{1}}$, that is $\Theta_{c-c_{1}}\left[\begin{array}{ccc} H_{c_{2}} & \cdots & H_{c_{I}} \end{array}\right]=0$, and ${\tilde{Z}}_{c-c_{1}}$ is the independent complex Gaussian random variable noise multiplied with the null-space matrix $\Theta_{c-c_{1}}$. The next step of MGSTC is to assume that
$${\tilde{Y}}_{c_{1}}=\sqrt{P}H_{c_{1}}^{\left(eff\right)}S_{c_{1}}+\sqrt{\frac{P}{SNR}}{\tilde{Z}}_{c-c_{1}}$$
where $H_{c_{1}}^{\left(eff\right)}:=\Theta_{c-c_{1}}H_{c_{1}}$, and then use it in order to compute the Maximum Likelihood (ML) approximation ${\tilde{S}}_{c_{1}}$ for $S_{c_{1}}$ as
$${\tilde{S}}_{c_{1}}=\frac{1}{\sqrt{P}}E\left[{\left(H_{c_{1}}^{\left(eff\right)}\right)}^{+}{\tilde{Y}}_{c_{1}}\right].$$

Obviously, the computation of ${\tilde{S}}_{c_{1}}$ from the very first step contains errors since it does not take into account the part $\sqrt{P}\Theta_{c-c_{1}}\Delta HS$. Note that the probability that $\Theta_{c-c_{1}}\Delta H=0$ is negligible. The interference part $\sqrt{P}\Theta_{c-c_{1}}\Delta HS$ that is received as an additive impact to the MGSTC decoding algorithm produces the most destructive effects, as it destroys the orthogonality of $\Theta_{c-c_{1}}$ relative to the other part of the MIMO channel–response matrix, a destruction that the MGSTC algorithm cannot deal with (see  [11]). Moreover, the classical solution of increasing the transmission power *P* [3,45] also increases the interference part and thus makes it a bad solution.

In this paper, we present a new method called the AP-HSIC, the generalized AP-HSIC, to deal with problems associated with interference estimation and digital interference cancellation in the presence of Rayleigh deep-fading effects. Our algorithm utilizes self-computational feedback at the receiver to decode any value of *M* in an M-QAM under challenging conditions of Rayleigh deep-fading channels. Additionally, we provide BER analysis and results for 16-, 32-, and 64-QAM for better understanding.

To calculate ΔH and *S* from Equation (Equation 6), we can initially use the standard model to estimate an approximate value for *S* based on the known channel–response matrix *H*. We then iterate Equation (Equation 6) to find the best approximations for ΔH and *S*, as described in Section 3. Below, we provide computational feedback detailing how to nullify ΔH by computing ΔH and utilizing it to accurately decode *S* and to update the new MIMO channel–response matrix to be H+ΔH without having to update the transmitter.

## 3. The Generalized AP-HSIC Algorithm for QAM Modulations

In a previous study [56], and relating to (Equation 9), an algorithm was proposed for the computational self-feedback at the receiver that identifies both unknown ΔH and an unknown symbol matrix $S\in PSK^{n\times k}$ that solve the following problem:(11)$$\begin{array}{cc} \; & \arg\min_{\Delta H,S}{∥Y-\left(H+\Delta H\right)S∥}_{F}^{2}\\ \; & \mathrm{such}\mathrm{that}:\;{∥S∥}_{F}\leq r_{max}\sqrt{nk}, \end{array}$$
where $r_{max}=\sqrt{2}$ is the maximum absolute value of the points in the PSK modulation. Since this problem is non-convex, and since $PSK^{n\times k}$ is a discrete finite set with at least two elements, the problem can be proven to be NP-complete. To circumvent the complexity of the problem, Ref. [56] adopted the following strategy. Assuming currently that ΔH is known, the minimal Frobenius-norm solution of (Equation 9) for *S* is given by
(12)$${\left(H+\Delta H\right)}^{+}Y=S,$$
which is linear in terms of the variables ${\left(H+\Delta H\right)}^{+}$ and *S*. Now, the solution space of the linear Equation (Equation 12) when ΔH is also unknown is given by
(13)$$\left\{\begin{array}{c} S=\left(W_{1}Y^{*}+W_{2}\right){\left(I_{m}+YY^{*}\right)}^{-1}Y\\ {\left(H+\Delta H\right)}^{+}=\left(W_{1}Y^{*}+W_{2}\right){\left(I_{m}+YY^{*}\right)}^{-1}, \end{array}\right.$$
where $W_{1}\in C^{n\times k}$ and $W_{2}\in C^{n\times m}$ are free parameters.

Finally, the following closed and convex sets in $C^{n\times m}\times C^{n\times k}$ were defined as follows:(14)$$\begin{array}{cc} \; & C_{0}=\left\{\begin{array}{c} \left({\left(H+\Delta H\right)}^{+},S\right)\left|\;\right.\Delta H\;\mathrm{arbitrary}\;\mathrm{and}\;S\in B\left(0,\sqrt{2nk}\right) \end{array}\right\}, \end{array}$$
and
(15)$$C_{1}=\left\{\left({\left(H+\Delta H\right)}^{+},S\right)\left|\;\right.\Delta H\;\mathrm{and}\;S\;\mathrm{are}\;\mathrm{given}\;\mathrm{by}\;\left(13\right)\right\}.$$

The algorithm finds *S* and ΔH, which solves Equation (Equation 12) by finding a point $\left({\left(H+\Delta H\right)}^{+},S\right)$ located in the intersection $C_{0}⋂C_{1}$. The algorithm finds a point of intersection by starting from an arbitrary point (however, in  [56], the more plausible starting point $\Delta H_{0}=0,S_{0}=H^{+}Y$ was chosen) and by projecting the points of an improvement onto the boundary of the convex sets (orthogonally) one at a time. Alternating between them leads to convergence (the algorithm was shown to converge to the point of intersection in a linear time, see  [56]). Finally, the algorithm projects the entries of *S* onto the closest points inside the modulation. The algorithm performed best because the matrices *S* with entries from the PSK modulations appear to be on the boundary of the ball $B\left(0,\sqrt{2nk}\right)$ so that any boundary point of $C_{0}⋂C_{1}$ was sufficient to deduce *S* closest to the modulation and ΔH satisfying (Equation 9) (note that this equation guarantees *S* with minimal Frobenius-norm for any given ΔH). This does not hold anymore when the entries of *S* are taken from QAM modulations since the modulation points have more than one radius of magnitude.

We, therefore, propose an alternative for the closed-convex set $C_{0}$, so the algorithm will also work with QAM modulations. The convex sets and the projections are similar to what was defined for PSK modulations. Given a QAM modulation, let $r_{max}$ be the maximum radius of magnitude of the modulation constellation. We define the ball of matrices $B\left(0,r_{max}\sqrt{nk}\right)$ as the set of all matrices *S* with the Frobenius-norm less than or equal to $r_{max}\sqrt{nk}$. Since for each $S\in QAM^{n\times k}$, ${∥S∥}_{F}=\sqrt{\sum_{i=1}^{n}\sum_{j=1}^{k}{\left|S_{i,j}\right|}^{2}}\leq r_{max}\sqrt{nk}$, then $S\in B\left(0,r_{max}\sqrt{nk}\right)$. Therefore, $B\left(0,r_{max}\sqrt{nk}\right)$ is a closed-convex set that contains all possible symbol matrices $S\in QAM^{n\times k}$. We therefore define the new closed-convex set $C_{0}$ as follows:(16)$$C_{0}=\left\{\begin{array}{c} \left({\left(H+\Delta H\right)}^{+},S\right)\left|\;\right.\Delta H\;\mathrm{arbitrary}\;\mathrm{and}\;S\in B\left(0,r_{max}\sqrt{nk}\right) \end{array}\right\}$$

In addition, let $P_{C_{0}}:\left(A,B\right)\to \left(A,\beta B\right)$ be the (orthogonal) projection function on the closed-convex set $C_{0}$, where
$$\beta =\left\{\begin{array}{c} 1\;\;\mathrm{if}\;\;{∥B∥}_{F}\leq r_{max}\sqrt{nk}\\ \frac{r_{max}\sqrt{nk}}{{∥B∥}_{F}}\;\;\mathrm{otherwise} \end{array}\right.$$

Finally, we define $P_{QAM}$ as a function that projects the entries of *S* onto the closest points of the modulation to calculate the final feasible *S*. For the definition of the orthogonal projection $P_{C_{1}}:\left(A,B\right)\to \left(\hat{A},\hat{B}\right)\in C_{1}$, we compute (see  [56] for the proof of correctness)
(17)$$\begin{array}{cc} \; & \left[\begin{array}{cc} W_{2} & W_{1} \end{array}\right]=\left[\begin{array}{cc} A\left(I_{m}+YY^{*}\right) & BY^{*}\left(I_{m}+YY^{*}\right) \end{array}\right]\cdot {\left[\begin{array}{cc} I_{m} & YY^{*}\\ Y^{*} & Y^{*}YY^{*} \end{array}\right]}^{+}. \end{array}$$
and set
(18)$$\left\{\begin{array}{c} \hat{A}=\left(W_{1}Y^{*}+W_{2}\right){\left(I_{m}+YY^{*}\right)}^{-1}\\ \hat{B}=\left(W_{1}Y^{*}+W_{2}\right){\left(I_{m}+YY^{*}\right)}^{-1}Y, \end{array}\right.$$
where $W_{1},W_{2}$ are given by (Equation 17).

 **Remark 1.**
*In  [56], the assumption that m≤n and H+ΔH is full rank was made, implying that $\left(H+\Delta H\right){\left(H+\Delta H\right)}^{+}=I_{m}$. The reasoning is as follows: $Y=\left(H+\Delta H\right)S$ with the genuine S and using the Moore–Penrose pseudo-inverse properties implies the following:*

$$\begin{array}{cc} & \left(H+\Delta H\right){\left(H+\Delta H\right)}^{+}Y\\ & =\left(H+\Delta H\right){\left(H+\Delta H\right)}^{+}\left(H+\Delta H\right)S\\ & =\left(H+\Delta H\right)S=Y. \end{array}$$

*Let $\tilde{S}={\left(H+\tilde{\Delta H}\right)}^{+}Y$ computed by Algorithm 1, where we may assume that $\tilde{\Delta H}=\Delta H$, since any destruction explaining the received signal would cancel its influence. Then,*

$$\left(H+\Delta H\right)\tilde{S}=\left(H+\Delta H\right){\left(H+\Delta H\right)}^{+}Y=Y,$$

*since by assumption $\left(H+\Delta H\right){\left(H+\Delta H\right)}^{+}=I_{m}$. Therefore, by solving ${\left(H+\Delta H\right)}^{+}Y=S$ instead of $Y=\left(H+\Delta H\right)S$, we do not lose connection with the genuine solution, and the algorithm might produce a relevant approximation.*


The AP-HSIC algorithm is given in Algorithm 1, where the procedures for $P_{C_{0}}$ and $P_{C_{1}}$ are given in Algorithm 2 and Algorithm 3, respectively. A flowchart of the algorithm explaining the flow of computations is given in Figure 1.
**Algorithm 1** AP-HSIC: Receiver Self-Feedback Algorithm For QAM**Require:** *An algorithm for computing Moore–Penrose pseudo-inverses and algorithms for computing $P_{C_{0}}$, $P_{C_{1}}$ and $P_{QAM}$.* **Input:** $H,Y,r_{max},\epsilon >0$. **Output:** ΔH and *S* such that ${\left(H+\Delta H\right)}^{+}Y=S$, where $S\in QAM^{n\times k}$.
1.$\Delta H_{0}\leftarrow 0$2.$S_{0}\leftarrow H^{+}Y$3.$\left({\left(H+\Delta H_{1}\right)}^{+},S_{1}\right)\leftarrow P_{C_{1}}\left({\left(H+\Delta H_{0}\right)}^{+},S_{0}\right)$4.$\left({\left(H+\Delta H_{1}\right)}^{+},S_{1}\right)\leftarrow P_{C_{0}}\left({\left(H+\Delta H_{1}\right)}^{+},S_{1}\right)$5.$\Delta H_{1}\leftarrow {\left({\left(H+\Delta H_{1}\right)}^{+}\right)}^{+}-H$6.t←17.**while** $\sqrt{{∥\Delta H_{t}-\Delta H_{t-1}∥}_{F}^{2}+{∥S_{t}-S_{t-1}∥}_{F}^{2}}>\epsilon$ **do**8.       $\begin{array}{c} \left({\left(H+\Delta H_{t+1}\right)}^{+},S_{t+1}\right)\leftarrow P_{C_{1}}\left({\left(H+\Delta H_{t}\right)}^{+},S_{t}\right) \end{array}$9.       $\begin{array}{c} \left({\left(H+\Delta H_{t+1}\right)}^{+},S_{t+1}\right)\leftarrow P_{C_{0}}\left({\left(H+\Delta H_{t+1}\right)}^{+},S_{t+1}\right) \end{array}$10.     $\Delta H_{t+1}\leftarrow {\left({\left(H+\Delta H_{t+1}\right)}^{+}\right)}^{+}-H$11.     t←t+112.**end while**13.$\Delta H\leftarrow \Delta H_{t}$14.$S\leftarrow P_{QAM}\left(S_{t}\right)$15.**return** t,ΔH,S

**Algorithm 2** The Orthogonal Projection $P_{C_{0}}$**Require:** *Matrix and Arithmetic Operations.* **Input:** $\left(A,B\right)$ such that *A* is n×m and *B* is n×k and $r_{max},n,k$. **Output:** $\left(\hat{A},\hat{B}\right)\in C_{0}$ the orthogonal projection of $\left(A,B\right)$ onto the boundary of $C_{0}$.
1.**if** ${∥B∥}_{F}>r_{max}\sqrt{nk}$ **then**2.     $\beta \leftarrow \frac{r_{max}\sqrt{nk}}{{∥B∥}_{F}}$3.**else**4.     β←15.**end if**6.$\left(\hat{A},\hat{B}\right)\leftarrow \left(A,\beta B\right)$7.**return** $\left(\hat{A},\hat{B}\right)$


**Algorithm 3** The Orthogonal Projection $P_{C_{1}}$**Require:** *An algorithm for computing Moore–Penrose pseudo-inverses.* **Input:** $\left(A,B\right)$ such that *A* is n×m and *B* is n×k and *Y*. **Output:** $\left(\hat{A},\hat{B}\right)\in C_{1}$ the orthogonal projection of $\left(A,B\right)$ onto the boundary of $C_{1}$.
1.$\begin{array}{c} \left[\begin{array}{cc} W_{2} & W_{1} \end{array}\right]\leftarrow \left[\begin{array}{cc} A\left(I_{m}+YY^{*}\right) & BY^{*}\left(I_{m}+YY^{*}\right) \end{array}\right]\cdot {\left[\begin{array}{cc} I_{m} & YY^{*}\\ Y^{*} & Y^{*}YY^{*} \end{array}\right]}^{+} \end{array}$2.$\hat{A}\leftarrow \left(W_{1}Y^{*}+W_{2}\right){\left(I_{m}+YY^{*}\right)}^{-1}$3.$\hat{B}\leftarrow \left(W_{1}Y^{*}+W_{2}\right){\left(I_{m}+YY^{*}\right)}^{-1}Y$4.**return** $\left(\hat{A},\hat{B}\right)$


The AP-HSIC assumptions are summarized as follows [7,15]:1.m≥n.2.k≥max(m,n)=m.3.$H_{t}:=H+\Delta H_{t}$ is full rank for any t≥0.

The first point is that, assuming CSIR only, the number of receiving antennas should be at least as large as the number of transmit antennas to achieve a high diversity order. Secondly, the number of frames (denoted as *k*) should be greater than the number of receiving antennas in order to increase the decoding rate. The paper  [56] rigorously demonstrates in Theorem F.2 on page 21 that the estimation error $E\left[{∥H_{\mathrm{estimated}}-H_{\mathrm{exact}}∥}_{F}^{2}\right]\to 0$ as k→∞. Furthermore, Theorem F.1 on page 21 conclusively proves that the subset of full-rank matrices in $C^{m\times n}$ is dense in $C^{m\times n}$. Therefore, for any given ϵ>0, a full-rank matrix $H_{t,\epsilon }$ exists, such that ${∥H_{t,\epsilon }-H_{t}∥}_{F}^{2}\leq \epsilon$, which allows the assumption that $H_{t}$ itself is full rank.

Regarding the complexity of the algorithm, let
$$\epsilon_{0}:=\sqrt{E\left[{∥S_{0}-S_{*}∥}_{F}^{2}+{∥\Delta H_{0}-\Delta H_{*}∥}_{F}^{2}\right]}$$
denote the starting error, where $S_{*}$ is the exact sent signal and $\Delta H_{*}$ is the exact interference that defines a global optimal solution. Then, in  [56], it was proved that the number of AP-HSIC iterations until the current error $\epsilon_{t}$ becomes less than or equal to the given error threshold ϵ is $O\left(\left|\ln\left(\frac{\epsilon }{\epsilon_{0}}\right)\right|\right)$. The complexity of each iteration is $O\left(\max{\left(m,n,k\right)}^{3}\right)=O\left(k^{3}\right)$ and is dominated by the pseudo-inverses calculations. Thus, the total complexity of the AP-HSIC algorithm is $O\left(k^{3}\cdot \left|\ln\left(\frac{\epsilon }{\epsilon_{0}}\right)\right|\right)$. In Theorem F.4, it was proven that, in the worst case, ϵ is bounded below as $\epsilon \geq \frac{n}{\sqrt{mSNR}}$. Thus, for a fixed number of transmit antennas *n* and for any given SNR, one should increase the number of receive antennas *m* accordingly, or for fixed *n* and *m*, one should increase the SNR accordingly to obtain the needed threshold error. However, our experience with the AP-HSIC algorithm shows that it converges quickly to a globally optimal solution for any practical range of n,m,SNR and ϵ>0 regardless of the choice of the initial point. This is due to the reduction of the problem (Equation 11) to a convex problem under the assumption of $H_{t},t\geq 0$ being full rank.

## 4. The LM-HSIC Algorithm

In this section, we will define a new algorithm that finds *S* and ΔH to solve the problem
(19)$$\begin{array}{cc} \; & \arg\min_{\Delta H,S}{∥Y-\left(H+\Delta H\right)S∥}_{F}^{2}\\ \; & \mathrm{such}\mathrm{that}:\;{∥S∥}_{F}=r_{av}\sqrt{nk}, \end{array}$$
where $r_{av}=E\left[\frac{{∥S∥}_{F}}{\sqrt{nk}}\right]$. The new algorithm uses Lagrange multipliers optimization. A significant advantage of this algorithm is its independence on the assumption that H+ΔH is full rank, unlike AP-HSIC, which requires this assumption. This allows for dealing with scenarios where ΔH is a interference receiver–response matrix that attempts to eliminate *H*, thus reducing the rank of H+ΔH. Therefore, the new algorithm is considered complementary to the AP-HSIC algorithm. Moreover, since it uses fewer Moore–Penrose pseudo-inverse computations per iteration, it performs better in terms of CPU time since it converges only locally and not globally. Let the Lagrangian be
(20)$$L\left(\Delta H,S,\lambda \right)={∥Y-\left(H+\Delta H\right)S∥}_{F}^{2}+\lambda \left({∥S∥}_{F}^{2}-r_{av}^{2}nk\right).$$

Let $U=\left(U_{1},U_{2},U_{3}\right)\in C^{m\times n}\times C^{n\times k}\times R$ be a directional matrix with ${∥U∥}_{F}=1$ and let h>0. Then, the directional derivative at $\left(\Delta H,S,\lambda \right)$ in the direction $U=\left(U_{1},0,0\right)$ with ${∥U∥}_{F}={∥U_{1}∥}_{F}=1$ is defined by
(21)$$\nabla_{U}L\left(\Delta H,S,\lambda \right)=\lim_{h\to 0^{+}}\frac{L\left(\Delta H+hU_{1},S,\lambda \right)-L\left(\Delta H,S,\lambda \right)}{h}.$$

We compute
$$\begin{array}{cc} & L\left(\Delta H+hU_{1},S,\lambda \right)-L\left(\Delta H,S,\lambda \right)\\ & ={∥Y-\left(H+\Delta H\right)S-hU_{1}S∥}_{F}^{2}-{∥Y-\left(H+\Delta H\right)S∥}_{F}^{2}\\ & =-2hℜ{\langle Y-\left(H+\Delta H\right)S,U_{1}S\rangle }_{F}+h^{2}{∥U_{1}S∥}_{F}. \end{array}$$

From which we conclude that
(22)$$\begin{array}{cc} \nabla_{U}L\left(\Delta H,S,\lambda \right) & =-2ℜ{\langle Y-\left(H+\Delta H\right)S,U_{1}S\rangle }_{F}\\ \; & =-2ℜ\left(trace\left(U_{1}^{*}\left(Y-\left(H+\Delta H\right)S\right)S^{*}\right)\right). \end{array}$$

To find a necessary condition for a minimum point of L, we set $\nabla_{U}L\left(\Delta H,S,\lambda \right)=0$, for any *U* as above. This implies that
(23)$$-2ℜ\left(trace\left(U_{1}^{*}\left(Y-\left(H+\Delta H\right)S\right)S^{*}\right)\right)=0.$$

Since the last holds true for any $U=\left(U_{1},0,0\right)$ as above, we conclude that
(24)$$\left(Y-\left(H+\Delta H\right)S\right)S^{*}=0.$$

Similarly, let $U=\left(0,U_{2},0\right)$ be a directional matrix with ${∥U∥}_{F}={∥U_{2}∥}_{F}=1$ and let h>0. Then, the directional derivative at $\left(\Delta H,S,\lambda \right)$ in the direction *U* is defined by
(25)$$\nabla_{U}L\left(\Delta H,S,\lambda \right)=\lim_{h\to 0^{+}}\frac{L\left(\Delta H,S+hU_{2},\lambda \right)-L\left(\Delta H,S,\lambda \right)}{h}.$$

We compute
$$\begin{array}{cc} & L\left(\Delta H,S+hU_{2},\lambda \right)-L\left(\Delta H,S,\lambda \right)={∥Y-\left(H+\Delta H\right)\left(S+hU_{2}\right)∥}_{F}^{2}\\ & \;\;\;\;+\lambda \left({∥S+hU_{2}∥}_{F}^{2}-r_{av}^{2}nk\right)-{∥Y-\left(H+\Delta H\right)S∥}_{F}^{2}-\lambda \left({∥S∥}_{F}^{2}-r_{av}^{2}nk\right)\\ & ={∥h\left(H+\Delta H\right)U_{2}∥}_{F}^{2}-2ℜ{\langle Y-\left(H+\Delta H\right)S,h\left(H+\Delta H\right)U_{2}\rangle }_{F}\\ & \;\;\;\;+{∥Y-\left(H+\Delta H\right)S∥}_{F}^{2}+\lambda {∥S∥}_{F}^{2}+2\lambda ℜ{\langle S,hU_{2}\rangle }_{F}\\ & \;\;\;\;+\lambda h^{2}{∥U_{2}∥}_{F}^{2}-\lambda r_{av}^{2}nk-{∥Y-\left(H+\Delta H\right)S∥}_{F}^{2}-\lambda \left({∥S∥}_{F}^{2}-r_{av}^{2}nk\right)\\ & =h^{2}{∥\left(H+\Delta H\right)U_{2}∥}_{F}^{2}-2hℜ{\langle Y-\left(H+\Delta H\right)S,\left(H+\Delta H\right)U_{2}\rangle }_{F}\\ & \;\;\;\;+\lambda \left(2hℜ{\langle S,U_{2}\rangle }_{F}+h^{2}{∥U_{2}∥}_{F}^{2}\right). \end{array}$$

From which we can conclude that
(26)$$\begin{array}{cc} \nabla_{U}L\left(\Delta H,S,\lambda \right)\; & =-2ℜ{\langle Y-\left(H+\Delta H\right)S,\left(H+\Delta H\right)U_{2}\rangle }_{F}+2\lambda ℜ{\langle S,U_{2}\rangle }_{F}\\ \; & =-2ℜ\left(trace\left(U_{2}^{*}\left({\left(H+\Delta H\right)}^{*}\left(Y-\left(H+\Delta H\right)S\right)-\lambda S\right)\right)\right). \end{array}$$

To find a necessary condition for a minimum point of L, we set $\nabla_{U}L\left(\Delta H,S,\lambda \right)=0$ for any $U=\left(0,U_{2},0\right)$ as above. This implies that
(27)$$\begin{array}{cc} \; & -2ℜ\left(trace\left(U_{2}^{*}\left({\left(H+\Delta H\right)}^{*}\left(Y-\left(H+\Delta H\right)S\right)-\lambda S\right)\right)\right)=0. \end{array}$$

Since the last holds true for any *U* as above, then
(28)$${\left(H+\Delta H\right)}^{*}\left(Y-\left(H+\Delta H\right)S\right)-\lambda S=0.$$

Finally, we calculate the last partial derivative for $U=\left(0,0,U_{3}\right)$ with ${∥U∥}_{F}=\left|U_{3}\right|=1$:(29)$$\begin{array}{cc} \; & \nabla_{U}L\left(\Delta H,S,\lambda \right)=U_{3}\left({∥S∥}_{F}^{2}-r_{av}^{2}nk\right)=0\;\;\Rightarrow {∥S∥}_{F}=r_{av}\sqrt{nk} \end{array}$$

From (Equation 24) and (Equation 28), we conclude
(30)$$\begin{array}{c} \lambda SS^{*}=0\Rightarrow \left|\lambda \right|{∥S∥}_{F}^{2}=0\Rightarrow \lambda =0, \end{array}$$
since ${∥S∥}_{F}^{2}=0$ would imply that $S=0_{n,k}$, which is impossible since 0∉QAM and since we assumed that ${∥S∥}_{F}=r_{av}\sqrt{nk}$.

Setting λ=0 into (Equation 28), we obtain
(31)$$\begin{array}{c} {\left(H+\Delta H\right)}^{*}\left(Y-\left(H+\Delta H\right)S\right)=0. \end{array}$$

Now, from (Equation 24), we obtain
(32)$$\begin{array}{c} YS^{*}=\left(H+\Delta H\right)SS^{*}, \end{array}$$
where, under the assumption that $SS^{*}$ is invertible, we obtain
(33)$$\begin{array}{c} YS^{*}{\left(SS^{*}\right)}^{-1}=H+\Delta H. \end{array}$$

From the properties of the Moore–Penrose pseudo-inverse, if *S* has a full row rank, then $S^{+}=S^{*}{\left(SS^{*}\right)}^{-1}$ and $SS^{+}=I_{n}$. However, even if $SS^{*}$ is not invertible, $YS^{+}-H$ yields the best approximation for ΔH in terms of the Frobenius-norm. This means that $\Delta H=YS^{+}-H$ has the minimal energy (with value ${∥\Delta H∥}_{F}^{2}={∥YS^{+}-H∥}_{F}^{2}$) that explains the current corrupted signal *Y*. There are two explanations for the assumption that ΔH has minimal energy. The first one is related to the fact that the noise intensity depends mainly on the adequate bandwidth of the communication system and the noise density, which depends on the ambient temperature and the Boltzmann constant. Many modern wireless communication systems with wide bandwidth are susceptible to this noise, which causes a significant decrease in the energy of the transmitted symbol to a minimum. In addition, it can be assumed that ΔH is part of the noise expression; therefore, taking the minimum energy of ΔH is a reasonable assumption [57]. The second explanation is related to the type of interrupter. Different kinds of jammers or interference phenomena can be destructive even with minimal energy; for example, an intelligent jammer or a deep-fade. The presence of these phenomena causes a change and rotation in the entire constellation received in the receiver and results in the process of accumulating destructive errors [11]. These reasons formed the guideline for choosing ΔH with minimal norm. We therefore set
(34)$$\begin{array}{c} \Delta H=YS^{+}-H, \end{array}$$
as the preferred solution for ΔH for known *S*.

From (Equation 28), we obtain
(35)$$\begin{array}{c} {\left(H+\Delta H\right)}^{*}Y={\left(H+\Delta H\right)}^{*}\left(H+\Delta H\right)S. \end{array}$$

As explained above, we obtain the following preferred solution for *S* when ΔH is known (even if H+ΔH is not full rank):(36)$$\begin{array}{c} {\left(H+\Delta H\right)}^{+}Y=S, \end{array}$$
as the best solution in terms of the Frobenius-norm.

To conclude, from (Equation 24), (Equation 28), and (Equation 29), we need to solve
(37)$$\left\{\begin{array}{c} \Delta H=YS^{+}-H\\ S={\left(H+\Delta H\right)}^{+}Y\\ {∥S∥}_{F}=r_{av}\sqrt{nk}, \end{array}\right.$$
to obtain a critical point of L for which ∇L=0.

 **Remark 2.**
*Note that (37) represents highly coupled non-linear non-convex equations of ΔH,S. Contrastingly, the function $f\left(\Delta H\right)={∥Y-\left(H+\Delta H\right)S∥}_{F}^{2}$ is a convex function of ΔH for a given S, and*

$$\begin{array}{cc} {∥Y-\left(H+\Delta H\right)S∥}_{F}^{2} & ={∥Y\left(I-S^{+}S\right)+\left(YS^{+}-\left(H+\Delta H\right)\right)S∥}_{F}^{2}\\ & ={∥Y\left(I-S^{+}S\right)∥}_{F}^{2}+{∥\left(YS^{+}-\left(H+\Delta H\right)\right)S∥}_{F}^{2}, \end{array}$$

*implying that the global minimum of f is accepted for ΔH satisfying*

$$\left(YS^{+}-\left(H+\Delta H\right)\right)S=0,$$

*which, by the assumption that S is full rank (and therefore $SS^{+}=I_{n}$), is equivalent to*

$$\Delta H=YS^{+}-H.$$

*Similarly, the function $g\left(S\right)={∥Y-\left(H+\Delta H\right)S∥}_{F}^{2}$ is a convex function of S for a given ΔH, and*

$$\begin{array}{cc} & {∥Y-\left(H+\Delta H\right)S∥}_{F}^{2}\\ & \;\;\;\;={∥\left(I-\left(H+\Delta H\right){\left(H+\Delta H\right)}^{+}\right)Y+\left(H+\Delta H\right)\left({\left(H+\Delta H\right)}^{+}Y-S\right)∥}_{F}^{2}\\ & \;\;\;\;={∥\left(I-\left(H+\Delta H\right){\left(H+\Delta H\right)}^{+}\right)Y∥}_{F}^{2}+{∥\left(H+\Delta H\right)\left({\left(H+\Delta H\right)}^{+}Y-S\right)∥}_{F}^{2}, \end{array}$$

*implying that the global minimum of g is accepted for S that minimizes*

$${∥\left(H+\Delta H\right)\left({\left(H+\Delta H\right)}^{+}Y-S\right)∥}_{F}^{2},$$

*that is, for*

$$S={\left(H+\Delta H\right)}^{+}Y.$$

*These are exactly the equations appearing in (37). This explains the fast convergence of the algorithm since once ΔH or S was identified correctly, the other is concluded immediately.*


Algorithm 4 presented below, which we call the Lagrange Multipliers Hard Successive Interference Cancellation (LM-HSIC), solves the problem (Equation 19), after which it uses $P_{QAM}$ to project the final *S* onto the closest point in the related $QAM^{n\times k}$ modulation. In its main loop, it computes $\Delta H_{t+1},S_{t+1}$ by calculating (37) as $YS_{t}^{+}-H$ and ${\left(H+\Delta H_{t}\right)}^{+}Y$, respectively, until the errors converge below some prescribed error threshold ϵ>0. The flow of computations is described in Figure 2.

The LM-HSIC assumptions are summarized as follows:1.k≥max(m,n)=m.2.$S_{t}$ is full rank for any t≥0.3.The exact sent signal satisfy ${∥S_{*}∥}_{F}\approx E\left[{∥S∥}_{F}\right]$.4.$\Delta H_{0},S_{0}$ is sufficiently close to the global minimum $\Delta H_{*},S_{*}$ of L, or at least $\Delta H_{0},S_{0}$ is sufficiently close to a local minimum of L.

The reasoning for k≥max(m,n) is the same as for the AP-HSIC. The reasoning for $S_{t}$ being full rank is to obtain an explicit solution for Equation (Equation 24) (see also Remark 2). However, note that the exact sent signal $S_{*}$ is n×k, where *k* depends on the receiver sample rate that can be chosen as k≥m≥n, thus allowing for $S_{*}$ to be full rank [7,15]. Moreover, the content of $S_{*}$ can be decided by protocols that guarantee it being full rank [7,15]. Thus, in this regard, LM-HSIC is less restrictive than the AP-HSIC.

The LM-HSIC algorithm’s complexity per iteration is $O\left(\max\left(m,n,k\right)\right)=O\left(k^{3}\right)$ due to pseudo-inverse computations. The number of iterations needed for convergence is uncertain, but typically, it converges in just a few iterations based on our experience.

A disadvantage of the algorithm is the assumption that the Frobenius-norm of the transmitted $S_{*}$ is close to $E\left[{∥S∥}_{F}\right]$. The problem with this assumption might arise in cases where ${∥S_{*}∥}_{F}$ is far from $E\left[{∥S∥}_{F}\right]$. For example, when $S_{*}=s_{0}1_{n\times k}$ (i.e., all the symbols are the same), where $s_{0}$ is a symbol in the modulation that has the minimal radius (or the maximal radius). Now, because of the constraint, the algorithm outputs $\tilde{S}$, where ${∥\tilde{S}∥}_{F}\approx E\left[{∥S∥}_{F}\right]$, and, therefore, the resulting $\tilde{S}$ will be inaccurate. This means that the algorithm will work best when the sent information is distributed between the radii of the QAM modulation points so that the radius of the sent signal $\frac{{∥S_{*}∥}_{F}}{\sqrt{nk}}$ is close to the expected radius $r_{av}=E\left[\frac{{∥S∥}_{F}}{\sqrt{nk}}\right]$, and the farther the radius is from the expected radius, the more errors the result will contain. Note, however, that $S_{*}=s_{0}1_{n\times k}$ and other “corner” points of $QAM^{n\times k}$ are very rare, if not impossible, in real-world applications, and can be avoided by protocols [7].
**Algorithm 4** LM-HSIC: Receiver Self-Feedback Algorithm For QAM**Require:** *An algorithm for computing Moore–Penrose Pseudo-inverses, and an algorithm for computing $P_{QAM}$.* **Input:** $H,Y,\epsilon >0,r_{av}=E\left[\frac{{∥S∥}_{F}}{\sqrt{nk}}\right],n,k$. **Output:** ΔH and *S* such that $Y=\left(H+\Delta H\right)S$, where $S\in QAM^{n\times k}$.
1.$\Delta H_{0}\leftarrow 0$2.$S_{0}\leftarrow H^{+}Y$3.$S_{0}\leftarrow \frac{S_{0}}{{∥S_{0}∥}_{F}}\cdot r_{av}\sqrt{nk}$4.$\Delta H_{1}\leftarrow YS_{0}^{+}-H$5.$S_{1}\leftarrow {\left(H+\Delta H_{1}\right)}^{+}Y$6.$S_{1}\leftarrow \frac{S_{1}}{{∥S_{1}∥}_{F}}\cdot r_{av}\sqrt{nk}$7.t←18.**while** $\left({∥\Delta H_{t}-\Delta H_{t-1}∥}_{F}>\epsilon \right)\vee \left({∥S_{t}-S_{t-1}∥}_{F}>\epsilon \right)$ **do**9.      t←t+110.     $\Delta H_{t}\leftarrow YS_{t-1}^{+}-H$11.     $S_{t}\leftarrow {\left(H+\Delta H_{t}\right)}^{+}Y$12.     $S_{t}\leftarrow \frac{S_{t}}{{∥S_{t}∥}_{F}}\cdot r_{av}\sqrt{nk}$13.**end while**14.$\Delta H\leftarrow \Delta H_{t}$15.$S\leftarrow P_{QAM}\left(S_{t}\right)$16.**return** t,ΔH,S


## 5. Simulations and Numerical Results

In order to demonstrate the effectiveness of the proposed AP-HSIC method and assess the performance of LM-HSIC, we conducted simulations of a communication environment using both the MATLAB and SIMULINK platforms. These simulations were based on three different methods described in Section 2, Section 3 and Section 4 with the same conditions and challenging scenarios. We compared the performance of these simulations using graphical and numerical aspects. Each simulation involves multiple stages that simulate various disturbance scenarios. We used the two architectures from the previous article [56] (with an extension to 16-, 32-, and 64-QAM modulation) and created a new architecture for LM-HSIC, as described below.

To enhance the architectures’ efficiency, the standard MGSTC transmitter encoder for all configurations integrates three OSTBC components. Each element is equipped with two transmission antennas, resulting in a total of six transmission antennas ($N_{t}=6$). On the receiving side, the signal matrix *Y* is captured through six receiving antennas ($N_{r}=6$). Furthermore, alongside the estimated MIMO channel matrix, the received signal is directed through the MGSTC decoding algorithm in the first architecture, the AP-HSIC algorithm in the second architecture, and the LM-HSIC algorithm in the third architecture. In the first architecture, the decoding process at the receiver is based on three Maximal-Ratio Combiners (MRC) components. We developed the environment of Rayleigh deep-fading using lotteries of the low values of the Rayleigh envelope (below 0.3), and we characterized the PDF of the SNR–Rayleigh distribution as a function of the SNR [58,59]:(38)$$f\left(SNR\right)=\frac{1}{\bar{SNR}}e^{-\frac{SNR}{\bar{SNR}}}$$

In addition, we ran the simulations based on the average SNR ($\bar{SNR}$).

In the first simulation, we used an MGSTC transmitter comprising an information generator block, symbol modulation block, and OSTBC-encoder blocks. The MIMO channel is quasi-static flat Rayleigh deep-fading and can change the AWGN intensity parameter. The receiver blocks consist of an MGSTC decoding algorithm and MRC blocks. Lastly, we included a BER calculator to measure the practical effects of the system’s performance. The second simulation also used the MGSTC transmitter and MIMO channel matrix block with the same components as described earlier. However, on the receiver side, instead of an MGSTC decoding algorithm block, we replaced it with an AP-HSIC block to strengthen the receiver’s ability to operate in parallel decoding communication mode. The third simulation was based on the same components described in the first and second simulations, but we changed the decoding algorithm from AP-HSIC to LM-HSIC.

We examined the systems’ architectures regarding the BER vs. average SNR by running the interference Rayleigh deep-fading scenario with high-level AWGN under three modulation orders as follows: 16-, 32-, and 64-QAM at the transmitter. Eventually, we compared the performances between MGSTC and AP-HSIC and between LM-HSIC and AP-HSIC.

The simulation performance parameters are determined by [60]. As it is widely recognized, the systematic BER of the MGSTC decoding algorithm always corresponds to the BER of its final iteration. The systematic BER of the MGSTC decoding algorithm was compared to the average total BER of the AP-HSIC and the average total BER of the LM-HSIC, as shown in the following figures.

### 5.1. MGSTC and AP-HSIC Simulations under Rayleigh Deep-Fading Scenario and High- Leveled AWGN

As described earlier, this section presents simulation results of the algorithms MGSTC and AP-HSIC in the case of deep-fading with high-leveled AWGN based on 16-QAM, 32-QAM, and 64-QAM modulations. In Figure 3, we present graphs showing the BER performance as an average SNR function using 16-QAM constellation modulation under the Rayleigh deep-fading scenario and high-leveled AWGN. These graphs compare the BER performance improvements of the AP-HSIC algorithm with the MGSTC algorithm. They demonstrate the BER performance of the three iterations of MGSTC alongside the overall BER performance of AP-HSIC. Similarly, the graphs at Figure 4a,b illustrate the graphs showing the BER performance of the MGSTC and the AP-HSIC decoding algorithms as an average SNR function under 32-QAM and 64-QAM constellation modulation, respectively, under the Rayleigh deep-fading scenario and high-leveled AWGN. In Figure 3 and Figure 4a,b, we observe that only the first iteration of the MGSTC decoding algorithm (only the 2×2 MIMO case) achieves a BER performance that enables decoding and extracting information. In contrast, the second and third iterations do not converge and exhibit high BER levels even at an extreme SNR. It can be seen that in Figure 3, the blue graph that represents the BER performance of the first iteration of the MGSTC scheme achieves $BER\approx 10^{-3}$ at an average SNR≈16 dB. The purple graph represents the BER performance of the AP-HSIC based on the full MIMO case 6×6 and achieves $BER\approx 10^{-3}$ at an average SNR≈14 dB. The yellow graph (which represents the BER performance of the second iteration of the MGSTC system based on the MIMO case 4×6) and the orange graph (which represents the total BER performance of the MGSTC system based on the full MIMO case 6×6) do not converge to BER levels that enable the reproduction of information. In Figure 4a, the blue graph representing the BER performance of the first iteration of the MGSTC decoding algorithm achieves $BER\approx 10^{-3}$ at an average SNR≈20 dB and achieves $BER\approx 5\cdot 10^{-4}$ only at an average SNR≈25 dB. The purple graph represents the BER performance of the AP-HSIC, achieves $BER\approx 10^{-3}$ at an average SNR≈19 dB, and achieves $BER\approx 10^{-6}$ at an average SNR≈25 dB. The yellow and orange graphs, representing the BER performance of the second and third iterations of the MGSTC, respectively, also do not converge to BER levels that enable the reproduction of information in the 32-QAM modulation. In Figure 4b, we can see that the blue graph represents the BER performance of the first iteration of the MGSTC scheme an, achieves $BER\approx 10^{-3}$ at an average SNR≈24 dB, and achieves $BER\approx 10^{-5}$ only at an average SNR≈30 dB in the 64-QAM modulation. The purple graph represents the BER performance of the AP-HSIC, achieves $BER\approx 10^{-3}$ at an average SNR≈20 dB, and achieves $BER\approx 10^{-6}$ at an average SNR≈28 dB in the 64-QAM modulation. The BER performance of the second and third iterations of the MGSTC, represented by the yellow and orange graphs, also fails to converge to BER levels that enable the reproduction of information in the 64-QAM modulation.

In our series of experiments, we have gained important insights and drawn conclusions. When comparing the BER performance of the AP-HSIC algorithm with the second and third iterations of the MGSTC decoding algorithm under scenarios of Rayleigh deep-fading in three types of modulation order, it is evident that the AP-HSIC algorithm outperforms the others. It exhibits superior BER performance and an efficient parallel decoding process. This is due to the fact that the second and third iterations of the MGSTC decoding algorithm fail to achieve the required BER performance for decoding the transmitted information (see Figure 5). The proposed AP-HSIC offers better diversity gain compared to the first iteration of the MGSTC algorithm, despite the fact that the transmission power of the first two antennas in the first iteration of the MGSTC is six times higher than in the full MIMO mode of the AP-HSIC scheme (6×6 in our proposal compared to 2×2 in the first iteration of the MGSTC scheme). In Figure 3, it can be observed that AP-HSIC achieves a diversity gain ≈2 dB higher compared to the first iteration of the MGSTC. In Figure 4a, the AP-HSIC presents a diversity gain ≈2 dB higher compared to the first iteration of the MGSTC. In Figure 4b, the AP-HSIC presents a diversity gain ≈5 dB higher compared to the first iteration of the MGSTC. In the proposed scheme, the BER performance is significantly improved compared to the first iteration of the MGSTC scheme. There is also an enhancement in spatial utilization, and, equally important, the energy efficiency of the AP-HSIC is significantly higher than that of MGSTC. Additionally, the ability of the AP-HSIC to mitigate interference when combined with parallel decoding (i.e., it allows parallel decoding and utilizes the operation of all six transmission/receiving antennas in parallel) is superior to the serial decoding and error accumulation of the MGSTC. The AP-HSIC scheme consistently shows lower BER levels at lower average SNR areas for all constellation orders than the MGSTC scheme.

Furthermore, it is worth noting that MGSTC has a significant limitation: errors accumulate with each iteration. This means that the accuracy of each iteration sets a lower bound for the accuracy of subsequent iterations. On the other hand, AP-HSIC, which relies on a parallel decoding process, consistently maintains robust BER performance across all decoding processes.

In Figure 5, we have conducted a detailed comparison of the systematic BER performance of MGSTC and AP-HSIC algorithms across different average SNR levels. We tested all three constellation orders under the Rayleigh deep-fading scenario and high-level AWGN. The systematic BER performance of the MGSTC algorithm does not converge to a suitable level for minimal normal decoding across all three scenarios. Conversely, the BER performance of the AP-HSIC system consistently converges to a level suitable for decoding in all three scenarios.

Figure 6 presents the number of iterations in the main loop of the AP-HSIC algorithm for each SNR value in 16-QAM modulation under the Rayleigh deep-fading scenario and high-leveled AWGN. Clearly, higher SNR values result in faster convergence. However, a more important and clearer conclusion is the fact that the value of SNR≈0 dB, the number of iterations is only five iterations, and, in SNR≈3 dB and higher, the number of iterations drops to only two iterations and stabilizes permanently. There are two significant explanations for this insight. The first explanation is that, even if the theoretical design of the AP-HSIC algorithm error, ϵ, in SNR≈3 dB, is ϵ≈0.0775 (e.g, $\epsilon \geq \frac{n}{\sqrt{mSNR}}$), practically, the actual algorithm error value drops drastically to a value of $\epsilon \approx 10^{-9}$. This is because activating a full MIMO array causes a high diversity gain. The second explanation is that, since the AP-HSIC samples several samples before the appearance of the disturber and keeps these samples in memory, a good starting point is created, which is expressed as $\Delta H_{0}=0$ and $S_{0}=H^{+}Y$. This starting point makes the algorithm converge very quickly to an optimal point (see also Remark 2).

In summary, Rayleigh deep-fading scenarios are extreme interference situations that cause the breakdown of the spatial SIC decoding algorithm, such as MGSTC. Increasing the transmission power does not seem to be effective in these interference scenarios; it actually amplifies the interference factors and significantly hampers the decoding processes, as indicated in (Equation 10). Our results demonstrate a substantial improvement in the performance of the AP-HSIC algorithm and its effective handling of interference scenarios such as Rayleigh deep-fading.

### 5.2. LM-HSIC and AP-HSIC Simulations under Rayleigh Deep-Fading and High-Leveled AWGN

In this section, we present simulation results for the algorithms LM-HSIC and AP-HSIC in the case of Rayleigh deep-fading and high-leveled AWGN based on 16-QAM, 32-QAM, and 64-QAM modulations. Figure 7 illustrates the detailed comparison of the BER performance of the algorithms AP-HSIC and LM-HSIC in the case of Rayleigh deep-fading and high-leveled AWGN scenarios based on 16-QAM, 32-QAM, and 64-QAM modulations. In this Figure, the continuous purple, black, and green graphs represent the BER performance of the AP-HSIC algorithm in 16-, 32-, and 64-QAM constellation order, respectively, while the dotted purple, black, and green graphs represent the BER performance of the LM-HSIC in 16-, 32-, and 64-QAM constellation order, respectively. It can be seen from Figure 7 that both the AP-HSIC and LM-HSIC attain comparable low BER values for each modulation. This means that both the AP-HSIC algorithm and the LM-HSIC algorithm converge in all three scenarios to a BER performance level that enables proper and accurate decoding. However, it is important to note that the AP-HSIC algorithm outperforms the LM-HSIC algorithm in terms of BER performance and achieves higher diversity gains across all three constellation orders. Furthermore, the graph of LM-HSIC shifts to the right relative to the graph of AP-HSIC, and this shift diminishes as the SNR increases. That is, both the AP-HSIC algorithm and the LM-HSIC algorithm converge in all three constellations to a BER performance level that enables a proper and accurate decoding process. However, it can be noticed that the BER performance of the AP-HSIC algorithm is better than the BER performance of LM-HSIC, and it achieves a higher diversity gain in all three constellation orders.

Figure 8 illustrates the number of iterations of both algorithms for each SNR value under 16-QAM modulation. We observe that the number of iterations of LM-HSIC remains constant throughout the graph, even when the SNR value is low. In contrast, AP-HSIC’s number of iterations increases as the SNR decreases. Hence, we conclude that the LM-HSIC algorithm runs faster than the AP-HSIC algorithm. However, this phenomenon is related to the LM-HSIC algorithm’s local convergence against the AP-HSIC algorithm’s global convergence.

In summary, in simulations with each of the modulations, the LM-HSIC algorithm has a better diversity gain than AP-HSIC, but the AP-HSIC’s performance is better. In addition, the LM-HSIC algorithm runs faster than AP-HSIC.

In Figure 9, we present comparisons between the BER performances of AP-HSIC under the scenarios of Rayleigh fading in the Single-Input Single-Output (SISO) case with 16-, 32-, 64-QAM to the theoretical 16-, 32-, and 64-QAM SISO AWGN only. We also present comparisons between the Rayleigh fading SISO case scenarios with 16-, 32-, and 64-QAM and no error correction algorithm. It is clear from Figure 9 that the numerical result of the AP-HSIC algorithm is situated between the trade-off region of the theoretical graphs to the system with no error correction algorithm. These results illustrate the reliability of the AP-HSIC algorithm and the decoding improvement capabilities in the parallel decoding mode under the assumptions of Rayleigh fading, including AWGN.

## 6. Discussion, Conclusions, and Future Directions

Dealing with interference offset demands precise spatial selectivity and accurate channel estimation through channel sharing and receiver–transmitter feedback. An intelligent algorithm upgrade for the receiver is crucial for achieving high interference cancellation capabilities and enabling cognitive and collaborative radio communications. This computational feedback is essential for advancing MIMO technology in modern wireless communications, especially for technologies relying on Ultra-Reliable Low-Latency Communication Coding (URLLC) [61], some of which must be able to respond instantaneously, such as robots, autonomous vehicles, and medical equipment. Moreover, some technologies physically do not allow feedback (e.g., satellite communications) [61]. Therefore, computational feedback is critical when applying these technologies.

The simulation results showed that there is a necessary balance between energy efficiency and computational capabilities. The findings suggest that merely boosting transmission power does not always decrease the BER when faced with different interference scenarios, especially when there is a requirement to enhance the amount of information and transmit at a higher data rate (such as when higher modulation is necessary). Equally important, the ability to decode a heavy information transmission in parallel spatial decoding mode and simultaneously to SIC in complex interference scenarios is critical for the next generation of modern wireless network MIMO systems. The diversity of SIC decoding algorithms, such as AP-HSIC and LM-HSIC, enables real-time decoding capabilities in the real world of advanced MIMO techniques and allows them to break through the limitations of real-time space-SIC algorithms.

In our study, we observed a significant improvement in decoding performance by analyzing the BER versus SNR graphs of AP-HSIC and LM-HSIC algorithms. We compared these algorithms to the decoding algorithm MGSTC when dealing with destructive interference phenomena such as Rayleigh deep-fading and high-level AWGN. Our theoretical analyses and simulations enabled us to highlight the main contributions of the AP-HSIC and LM-HSIC decoding algorithms, which are as follows: Improved BER versus SNR performance, resulting in significantly enhanced diversity gain and diversity order of the AP-HSIC and the LM-HSIC compared to MGSTC, especially in relation to the second and third iterations of MGSTC, which fail to converge to a minimum BER for information decoding. The AP-HSIC and LM-HSIC algorithms play a crucial role in enhancing energy efficiency. A more efficient communication system combining the AP-HSIC and LM-HSIC decoding algorithms, allowing for the transmission of six times less transmission energy per symbol compared to the first iteration of MGSTC (which has an order antenna array of 2×6), three times less transmission energy per symbol compared to the second iteration (order antenna array of 4×6), and 1.5 times less transmission energy per symbol compared to the third iteration (order antenna array of 6×6). This is achieved while maintaining a fixed-order antenna array of 6×6 antennas in the AP-HSIC and LM-HSIC without needing orthonormal matrix bases to mitigate interferences, which are ineffective in Rayleigh deep-fading scenarios. Additional contributions are the ability of AP-HSIC and LM-HSIC algorithms to decode information in parallel compared to the serial decoding of MGSTC, as well as the capability of computational feedback without the need for closed-loop MIMO capabilities, enabling the handling of diverse interference scenarios such as Rayleigh deep-fading.

In our upcoming research, we aim to enhance the concept of computational feedback, examine assumptions regarding selective channels in frequency and time-space, and explore non-stationary channels in a comprehensive manner. We also intend to develop techniques (based on Machine Learning) to estimate the channels between the transmitter and the receiver and evaluate the channels between the interference and the receiver, including identifying and classifying the interference or jamming. Our overall aim is to produce optimal feedback that offsets interference and increases the channel capacity and reliability under the same concept of optimal feedback. 

## Figures and Tables

**Figure 1 sensors-24-05038-f001:**
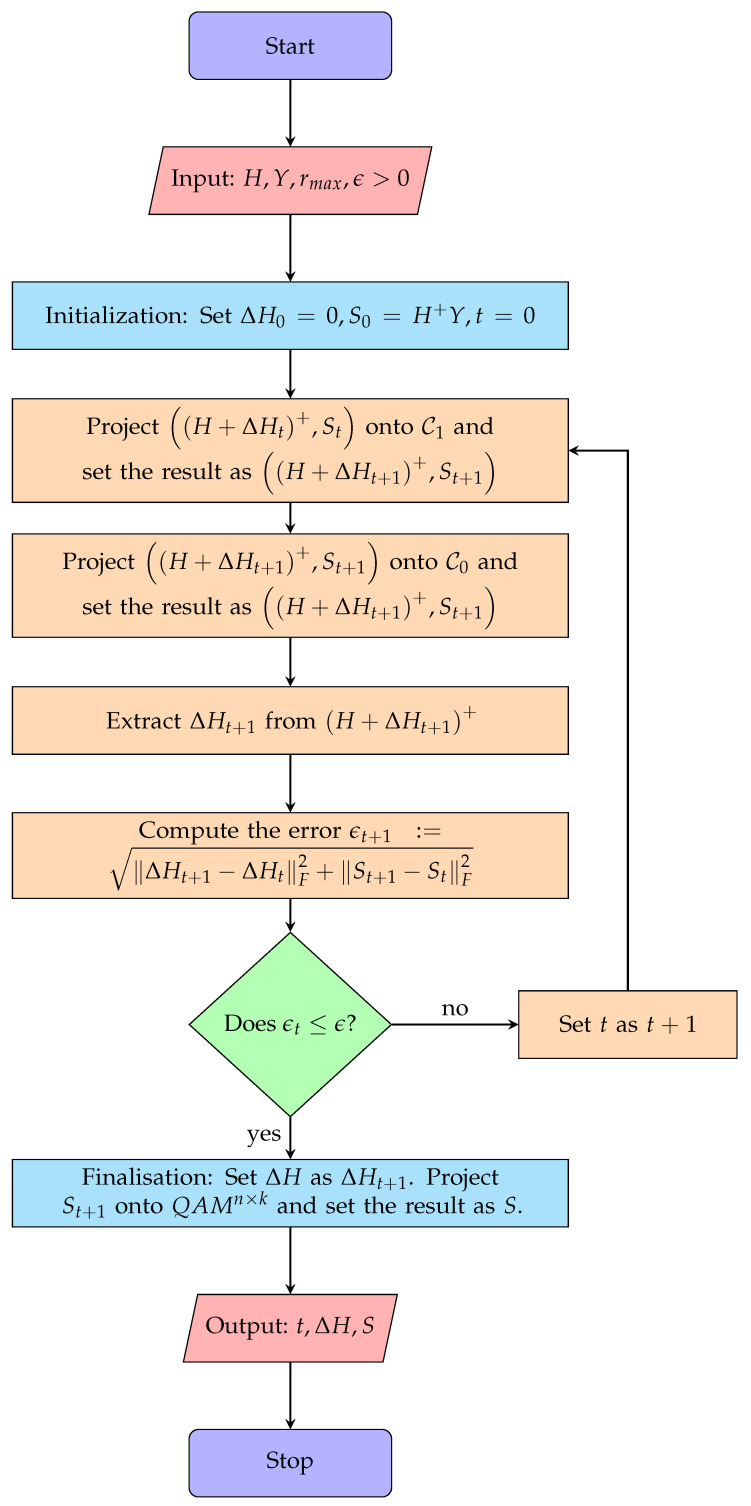
AP-HSIC algorithm flowchart.

**Figure 2 sensors-24-05038-f002:**
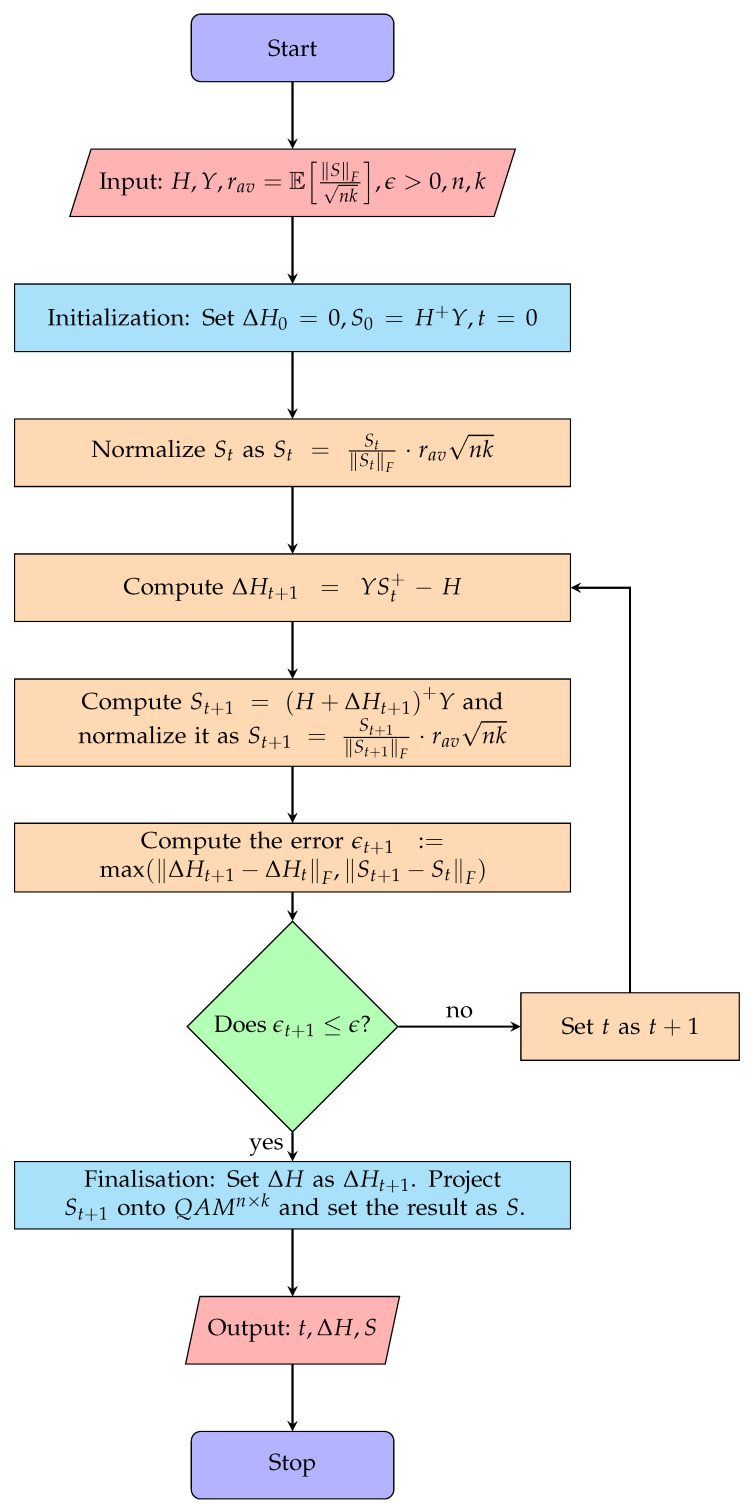
LM-HSIC algorithm flowchart.

**Figure 3 sensors-24-05038-f003:**
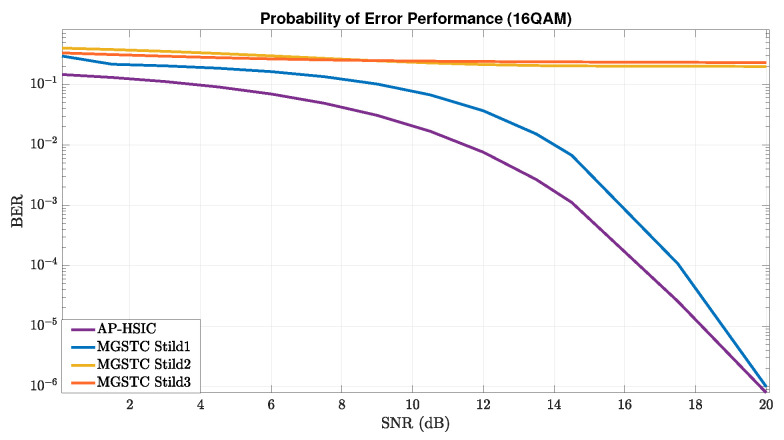
Plot graphs of BER performance vs. the average SNR that compares three iterations of the MGSTC algorithm and systematic iteration of the AP-HSIC algorithm under 16-QAM modulation in the presence of Rayleigh deep-fading and high-level AWGN.

**Figure 4 sensors-24-05038-f004:**
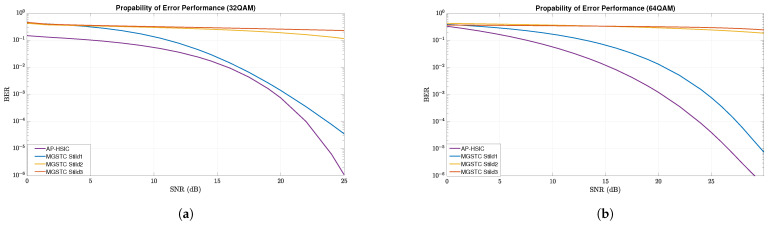
BER performance vs. the average SNR of three iterations of the MGSTC algorithm and systematic iterations of the AP-HSIC algorithm. (**a**) Plot graphs of BER performance vs. the average SNR that compares three iterations of the MGSTC algorithm and systematic iterations of the AP-HSIC algorithm under 32-QAM modulation in the presence of Rayleigh deep-fading and high-level AWGN. (**b**) Plot graphs of BER performance vs. the average SNR that compares three iterations of the MGSTC algorithm and systematic iterations of the AP-HSIC algorithm under 64-QAM modulation in the presence of Rayleigh deep-fading and high-level AWGN.

**Figure 5 sensors-24-05038-f005:**
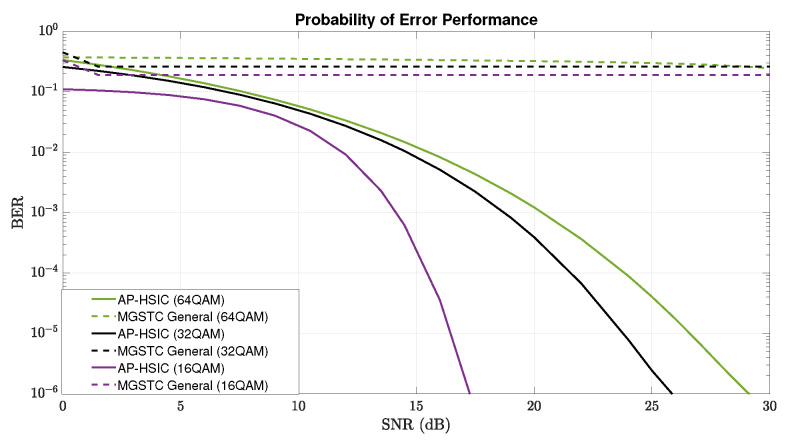
Plot graphs of BER performance vs. the average SNR for three systematic iterations using the MGSTC algorithm and AP-HSIC algorithm. Each iteration is represented by 16-QAM, 32-QAM, and 64-QAM modulations, respectively. The performance was compared under the presence of Rayleigh deep-fading and high-level AWGN.

**Figure 6 sensors-24-05038-f006:**
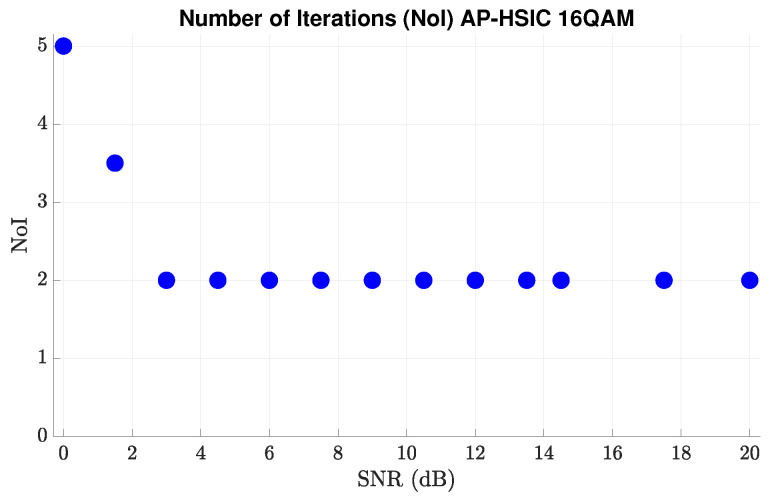
Average number of iterations for each average SNR value of AP-HSIC under 16-QAM modulation in the presence of Rayleigh deep-fading and high-level AWGN.

**Figure 7 sensors-24-05038-f007:**
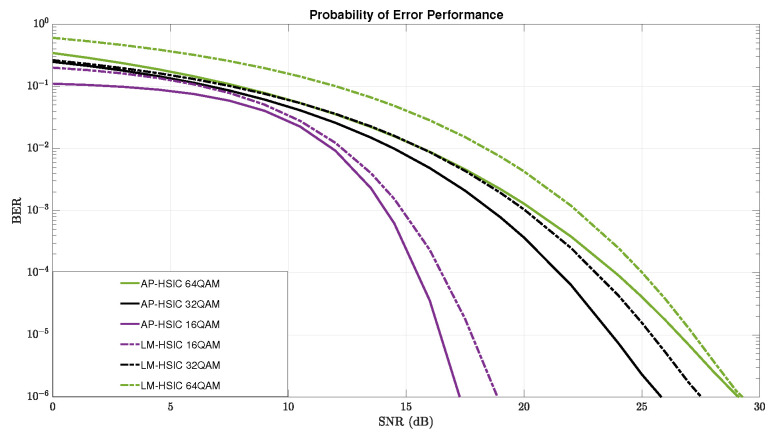
Plot graphs of BER performance vs. the average SNR for three systematic iterations using the LM-HSIC algorithm and AP-HSIC algorithm. Each iteration is represented by 16-QAM, 32-QAM, and 64-QAM modulations, respectively. The performance was compared under the presence of Rayleigh deep-fading and high-level AWGN.

**Figure 8 sensors-24-05038-f008:**
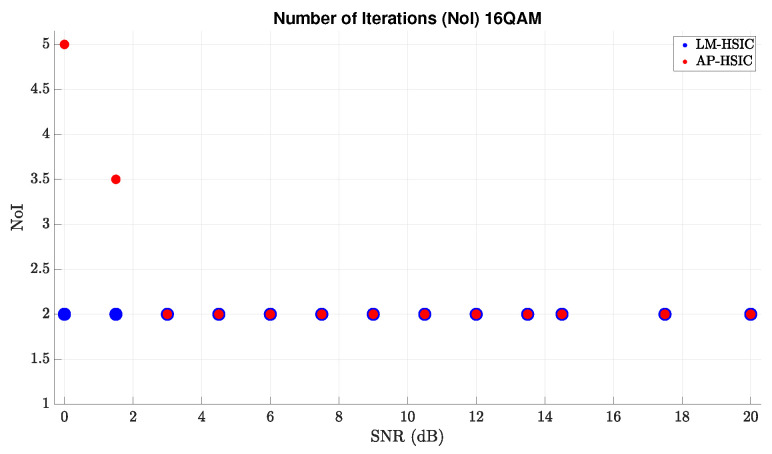
Comparisons of an average number of iterations for each average SNR value between LM-HSIC and AP-HSIC under 16-QAM modulation in the presence of Rayleigh deep-fading.

**Figure 9 sensors-24-05038-f009:**
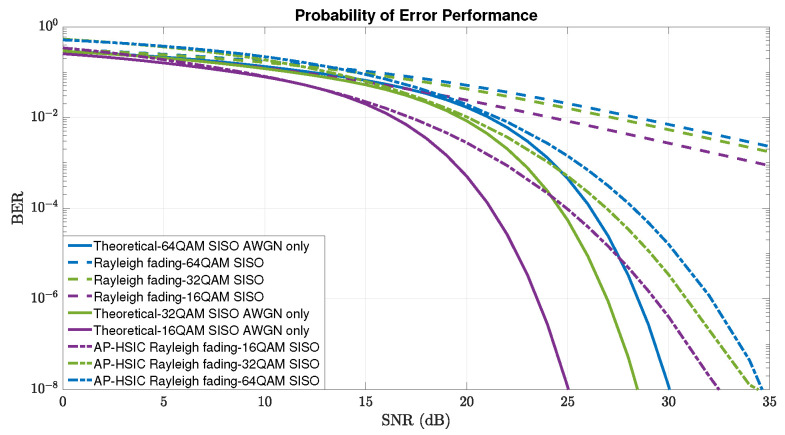
Comparisons between BER performances vs. the average SNR of AP-HSIC under the scenarios of Rayleigh fading–SISO with 16-, 32-, and 64-QAM and the theoretical 16-, 32-, and 64-QAM SISO AWGN only, and between the scenarios of Rayleigh fading SISO 16-, 32-, and 64-QAM without any correction algorithm.

## Data Availability

The Matlab/Simulink code files are available upon request from the authors.

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
