# Peer review of "Hard Successive Interference Cancellation for M-QAM MIMO Links in the Presence of Rayleigh Deep-Fading"

_sensors, 2024, doi:10.3390/s24155038_

Round 1
Reviewer 1 Report
Comments and Suggestions for Authors
The paper introduces an advanced digital interference cancellation algorithm, AP-HSIC, designed to decode M-QAM systems under Rayleigh deep-fading MIMO channels with high-leveled AWGN. The authors claim that the algorithm is effective, fast, and integrates well with LPCOD techniques, including high-order OSTBC. The paper also presents a comparison with the MGSTC decoding algorithm and the Lagrange Multipliers Hard Successive Interference Cancellation (LM-HSIC) algorithm, showcasing the AP-HSIC's superiority in terms of BER performance. However, I have the following comments regarding this paper.
1. The technical aspects of the paper are generally sound, with clear explanations of the proposed algorithm and its mathematical foundations. However, the paper could benefit from additional clarity on the assumptions made during the development of the algorithm and how these assumptions might impact its performance in real-world scenarios.
2. It would be beneficial to include a deeper analysis of the computational complexity and the scalability of the proposed algorithm for varying sizes of MIMO systems, which will provide a clearer picture of its practical applicability.
3. The figures presented in the paper effectively illustrate the performance comparison between the algorithms. However, the inclusion of more visual aids, such as flowcharts or diagrams, to better explain the algorithm's flow and decision points could enhance the reader's understanding.
4. The authors have provided a detailed literature review highlighting the importance and challenges of MIMO communication in the presence of various interferences and deep-fading scenarios. However, the review could be further strengthened by including more recent studies that address similar challenges with different methodologies, which would provide a more comprehensive background and positioning of the proposed AP-HSIC algorithm. Machine Learning-enabled MIMO-FBMC Communication Channel Parameter Estimation in IIoT: A Distributed CS Approach. Digital Communications and Networks, 2023.
The paper is of high quality and offers significant contributions to the field of MIMO communication systems. However, based on the points mentioned above, it is recommended that the paper undergoes major revisions to address the suggested areas of improvement. These revisions will help in elevating the paper further and making it a more impactful contribution to the field.
Comments on the Quality of English LanguageEnglish writing is OK.
Author Response
28/07/2024
Dear Ms. Sirikorn Suriyawong
Assistant Editor,
We are writing you in regards with our paper entitled:
"Hard successive interference cancellation for M-QAM MIMO link in the presence of Rayleigh deep-fading" (Manuscript ID sensors-3123250).
We first wish to thank you for finding and convening qualified reviewers to examine our paper. We would also like to thank you very much for your professional and quick handling of our article.
We thank the reviewers for their encouraging and professional responses that help us to improve our article.
We accept all the comments of the respected reviewers and fixed all the comments point by point in accordance of their remarks.
We believe that the readers of the MDPI-Sensors will be benefited from this elaborate study
Sincerely,
The authors.
Reviewer 1
The paper introduces an advanced digital interference cancellation algorithm, AP-HSIC, designed to decode M-QAM systems under Rayleigh deep-fading MIMO channels with high-leveled AWGN. The authors claim that the algorithm is effective, fast, and integrates well with LPCOD techniques, including high-order OSTBC. The paper also presents a comparison with the MGSTC decoding algorithm and the Lagrange Multipliers Hard Successive Interference Cancellation (LM-HSIC) algorithm, showcasing the AP-HSIC's superiority in terms of BER performance. However, I have the following comments regarding this paper.
1. The technical aspects of the paper are generally sound, with clear explanations of the proposed algorithm and its mathematical foundations. However, the paper could benefit from additional clarity on the assumptions made during the development of the algorithm and how these assumptions might impact its performance in real-world scenarios.
Following this important remark we elaborated the description of the algorithm. Note that both the AP-HSIC and LM-HSIC algorithms are now supplemented with a list of assumptions and explanations based on mathematics and communication principles. All our assumptions are in line with the requirements of MIMO communications. For example, in a CSIR-only scenario, in order to achieve high diversity gain, the number of receiver antennas needs to be at least equal to the number of transmitter antennas. Additionally, the number of frames k should be significantly higher than the number of receiver/transmitter antennas to attain a higher upper bound on the channel capacity. Also, as k increases, decoding errors and channel matrix estimation errors tend to zero.
In the AP-HSIC algorithm, the mathematical assumption that H+ΔH is full-rank is crucial for turning the non-linear non-convex problem into a convex one that can be solved efficiently. However, in real-world communication systems, the assumption happens to be true, as random matrices are full-rank with very high probability.
In the scenarios of a distortive jammer where ΔH is in the direction of –H, a drop in rank will appear. In that case, we propose using the LM-HSIC algorithm, which does not require this assumption.
The LM-HSIC assumptions require that S be full-rank. This condition can be satisfied by choosing k appropriately and using communication protocols to ensure that S is indeed full-rank. In communication practice, it is common to ensure theoretical properties by using communication protocols or standards. With regard to LM-HSIC, certain theoretical assumptions are made to guarantee its convergence at least to a local minimum of the Lagrangian. Currently, we do not have proof of convergence and rely on its practical performance, which demonstrates a very fast convergence rate. Although the underlying function is non-linear non-convex and thus makes the problem hard, a remark has been made, proving that the underlying target function is convex separately on ΔH (when S is held constant) and on S (when ΔH is held constant), for which we have exact optimal solutions. This lends support to the observed fast convergence in practice.
In accordance with the above, we extend the description of the algorithm, correspondigally.
2. It would be beneficial to include a deeper analysis of the computational complexity and the scalability of the proposed algorithm for varying sizes of MIMO systems, which will provide a clearer picture of its practical applicability.
In accordance with this remark, we evaluated the complexity of the AP-HSIC algorithm, considering the complexity of each iteration. This complexity is dependent on the sizes m, n, and k of the communication system, as well as the number of iterations required for convergence (i.e., until the error drops below the given threshold). Additionally, we have established a lower bound on the achievable error based on the number of transmitter antennas (n), which is inversely dependent on the square root of the number of receiver antennas (m) and inversely dependent on the square root of the SNR level. These findings are based on our previous research.
Currently, we do not have an upper bound on the number of iterations or a lower bound on the achievable error for the LM-HSIC. We rely on its practical performance. Simulation results demonstrate the efficacy of both algorithms across a wide range of SNR values and various interference scenarios including Rayleigh deep-fading scenarios, commonly found in practical situations. Our focus was aligned with the vision of next-generation communication systems, particularly with regard to interference effects prevalent in indoor applications and urban regions.
We updated this information in the revised manuscript.
3. The figures presented in the paper effectively illustrate the performance comparison between the algorithms. However, the inclusion of more visual aids, such as flowcharts or diagrams, to better explain the algorithm's flow and decision points could enhance the reader's understanding.
We have included flow charts to better illustrate the algorithms' computation flow as requested.
4. The authors have provided a detailed literature review highlighting the importance and challenges of MIMO communication in the presence of various interferences and deep-fading scenarios. However, the review could be further strengthened by including more recent studies that address similar challenges with different methodologies, which would provide a more comprehensive background and positioning of the proposed AP-HSIC algorithm. Machine Learning-enabled MIMO-FBMC Communication Channel Parameter Estimation in IIoT: A Distributed CS Approach. Digital Communications and Networks, 2023.
In accordance with this proposal, Citations have been added to include studies that address the effects of wireless channels and address information bottleneck issues. Destructive phenomena in the communication medium require the solution of optimization problems that involve choosing between conflicting demands. For example, increasing the transmission power leads to increased interference in the presence of reflective surfaces or objects. Additionally, citations addressing classical and machine-learning methodologies that deal with diffractive effects in wireless channels have been included. In addition, we strengthened the article with an important quote on the subject : " Machine Learning-enabled MIMO-FBMC Communication Channel Parameter Estimation in IIoT".
The paper is of high quality and offers significant contributions to the field of MIMO communication systems. However, based on the points mentioned above, it is recommended that the paper undergoes major revisions to address the suggested areas of improvement. These revisions will help in elevating the paper further and making it a more impactful contribution to the field.
Thank you for your positive perspective, valuable suggestions, and constructive feedback, which have certainly improved the paper.

Reviewer 2 Report
Comments and Suggestions for Authors
The purpose of the article is to introduce and analyze the Alternating Projections Digital Hard-Successive-Interference-Cancellation (AP-HSIC) algorithm for decoding M-QAM signals in MIMO systems under Rayleigh deep-fading conditions. It aims to demonstrate the effectiveness of this algorithm in handling high-level AWGN and various interference phenomena without requiring closed-loop feedback. The study compares AP-HSIC's performance against traditional MGSTC and the newly proposed LM-HSIC algorithms, focusing on bit error rate (BER) performance. The research intends to highlight the advantages of AP-HSIC in enhancing decoding accuracy and interference cancellation in complex wireless communication scenarios.
Major comments:
1)Correct the title capitalization and ensure it is consistent with the journal's style guide. For example, "Hard-Successive-Interference-Cancellation for M-QAM MIMO Link in the Presence of Rayleigh Deep-Fading" should be properly formatted.
2)The abstract should be more concise. It currently repeats concepts and lacks clarity in explaining the key contributions and findings. For instance, simplify the explanation of the AP-HSIC algorithm's integration with LPCOD techniques and its advantages.
3)The introduction has redundant information. For instance, the phrases "Advanced wireless MIMO communication technology is developing rapidly" and "considerable research attention" are repetitive. Streamline to focus on the problem statement and the study's motivation.
4)Clarify the significance of the study in the broader context of wireless communication technology, perhaps by including recent advancements or specific applications (e.g., 5G, IoT).
5)Equations should be numbered consecutively and referenced correctly in the text. For instance, the equation "(H + ∆H)S + Z" should have a clear reference in the discussion following it.
6)The algorithm descriptions (AP-HSIC and LM-HSIC) are dense and could benefit from flowcharts or diagrams for better readability and understanding. Also, ensure that the step-by-step process is clearly presented, possibly using bullet points or sub-sections.
7)Ensure all figures and graphs are labeled clearly and referenced correctly in the text. For example, "Figure 1" should be explicitly described in the body text to highlight its significance and findings.
8)The presentation of simulation results can be enhanced by providing more comparative analyses and discussions on the implications of the findings. For instance, explicitly state why AP-HSIC outperforms MGSTC under specific conditions.
9)The conclusions section should succinctly summarize the key findings and their implications. Remove any redundant statements and focus on the contributions of the study and potential future work.
Minor comments:
1)The citation details are incomplete. Ensure that the journal name, volume, issue, page numbers, and DOI are provided where currently missing.
2)Ensure the keywords are formatted consistently. "MIMO, deep-fading, interference-cancellation, parallel-decoding, computational-feedback" should be separated by semicolons and properly capitalized if required by the journal.
3)Ensure that all technical terms and acronyms are defined upon first use. For example, clarify terms like "AWGN," "OSTBC," and "DSSA" immediately when they are first mentioned.
4)Add captions that explain what each figure represents, ensuring they are self-explanatory without needing to refer back to the text.
5)The references should follow a consistent citation style. Ensure all cited works are included in the reference list and formatted according to the journal's guidelines.
Comments on the Quality of English LanguageImprove language clarity and grammar throughout the article. For instance, the phrase "creating self-interference between the antenna array" could be revised to "causing self-interference within the antenna array."
Avoid overly complex sentences that may hinder readability.
Extensive editing of the English language is required.
Author Response
28/07/2024
Dear Ms. Sirikorn Suriyawong
Assistant Editor,
We are writing you in regards with our paper entitled:
"Hard successive interference cancellation for M-QAM MIMO link in the presence of Rayleigh deep-fading" (Manuscript ID sensors-3123250).
We first wish to thank you for finding and convening qualified reviewers to examine our paper. We would also like to thank you very much for your professional and quick handling of our article.
We thank the reviewers for the encouraging and professional responses that help us to improve our article.
We accept all the comments of the respected reviewers and fixed all the comments point by point in accordance of their remarks.
We believe that the readers of the MDPI-Sensors will be benefited from the elaborate study.
Sincerely,
The authors.
Review 2
The purpose of the article is to introduce and analyze the Alternating Projections Digital Hard-Successive-Interference-Cancellation (AP-HSIC) algorithm for decoding M-QAM signals in MIMO systems under Rayleigh deep-fading conditions. It aims to demonstrate the effectiveness of this algorithm in handling high-level AWGN and various interference phenomena without requiring closed-loop feedback. The study compares AP-HSIC's performance against traditional MGSTC and the newly proposed LM-HSIC algorithms, focusing on bit error rate (BER) performance. The research intends to highlight the advantages of AP-HSIC in enhancing decoding accuracy and interference cancellation in complex wireless communication scenarios.
Thank you for your positive vision and helpful comments. They have greatly contributed to improving our article.
Major comments:
Correct the title capitalization and ensure it is consistent with the journal's style guide. For example, "Hard-Successive-Interference-Cancellation for M-QAM MIMO Link in the Presence of Rayleigh Deep-Fading" should be properly formatted.
The title was fixed according to the journal's style guide.
The abstract should be more concise. It currently repeats concepts and lacks clarity in explaining the key contributions and findings. For instance, simplify the explanation of the AP-HSIC algorithm's integration with LPCOD techniques and its advantages.
Following this important remark, the abstract has been edited and rephrased to improve clarity and conciseness while highlighting the major contributions and findings.
The introduction has redundant information. For instance, the phrases "Advanced wireless MIMO communication technology is developing rapidly" and "considerable research attention" are repetitive. Streamline to focus on the problem statement and the study's motivation.
Based on this feedback, we have eliminated all redundancies and improved the clarity of the introduction. It now highlights the importance of the problem for the advancement of practical communication systems, the motivation to address the problem, and a comprehensive literature review on the problem and its solution. We have also thoroughly examined the assumptions of current solutions and emphasized the need for the proposed solutions in practical communication systems, especially in light of the rapidly advancing next-generation wireless communication technology. With this context in mind, we have underscored the necessity of the proposed solutions in practical communication systems.
Clarify the significance of the study in the broader context of wireless communication technology, perhaps by including recent advancements or specific applications (e.g., 5G, IoT).
Following this important remark. We added a new paragraph (lines 54-81) describing the potential integration of the proposed algorithms with wireless communication technology in general, and refer also to 5G, IIoT, 6G, networks., including to the the 5G-NR HetNet's radio intelligent control mechanism.
Equations should be numbered consecutively and referenced correctly in the text. For instance, the equation "(H + ∆H)S + Z" should have a clear reference in the discussion following it.
The numbering of equations and equation references within the text has been corrected.
The algorithm descriptions (AP-HSIC and LM-HSIC) are dense and could benefit from flowcharts or diagrams for better readability and understanding. Also, ensure that the step-by-step process is clearly presented, possibly using bullet points or sub-sections.
Thank you for your proposal. We have included flow charts to better illustrate the algorithms' computation flow.
Ensure all figures and graphs are labeled clearly and referenced correctly in the text. For example, "Figure 1" should be explicitly described in the body text to highlight its significance and findings.
All figures and graphs have been corrected following your instructions. They are clearly marked and correctly referenced in the text.
The presentation of simulation results can be enhanced by providing more comparative analyses and discussions on the implications of the findings. For instance, explicitly state why AP-HSIC outperforms MGSTC under specific conditions.
Thanks for the feedback and constructive criticism. At your request, we have made substantial improvements to the simulations and numerical results chapter by expanding the comparative analyzes and discussions on the implications of the findings.
These changes are in the improved version between lines 555-567, 571-640 and 649-669.
The conclusions section should succinctly summarize the key findings and their implications. Remove any redundant statements and focus on the contributions of the study and potential future work.
In accordance with this remark, We have revised the conclusions chapter by removing unnecessary statements, expanding on the research contributions, and providing a brief summary of the main findings. You can see these changes between lines 690-704 and lines 711-735.
Minor comments:
The citation details are incomplete. Ensure that the journal name, volume, issue, page numbers, and DOI are provided where currently missing.
Following this comment, the details have been fully edited one by one.
Ensure the keywords are formatted consistently. "MIMO, deep-fading, interference-cancellation, parallel-decoding, computational-feedback" should be separated by semicolons and properly capitalized if required by the journal.
Your comments have been fully edited one by one.
Ensure that all technical terms and acronyms are defined upon first use. For example, clarify terms like "AWGN," "OSTBC," and "DSSA" immediately when they are first mentioned.
Your comments have been fully edited one by one.
Add captions that explain what each figure represents, ensuring they are self-explanatory without needing to refer back to the text.
DONE !
The references should follow a consistent citation style. Ensure all cited works are included in the reference list and formatted according to the journal's guidelines.
The process of changing the citation style in LaeTex is very complicated, and from our past experience, MDPI editing services can perform this more easily . In order to meet the review deadline, we preferred not to get involved with changes in the citation style in this stage and ask for MDPI editing services help.
Comments on the Quality of English Language
Improve language clarity and grammar throughout the article. For instance, the phrase "creating self-interference between the antenna array" could be revised to "causing self-interference within the antenna array."
Avoid overly complex sentences that may hinder readability.
Extensive editing of the English language is required.
Thank you for your positive perspective, valuable suggestions, and constructive feedback, which have certainly improved the paper.
We have made significant improvements to the article based on your guidance and constructive comments. We extensively edited the English language, enhancing the overall writing quality.
Additionally, we simplified complex sentences, added more sources to strengthen the article, and refined the research goals and its unique contributions to modern wireless communication while focusing on the main points.

Round 2
Reviewer 1 Report
Comments and Suggestions for Authors
The authors had carefully revised the manuscript, i think it can be accepted.
Reviewer 2 Report
Comments and Suggestions for Authors
I have no additional remarks in the revised version
Thanks
Comments on the Quality of English LanguageMinor editing of the English language is required.